# The Best of Both Worlds: On the Dilemma of Out-of-distribution Detection

**Qingyang Zhang[1]***, **Qiuxuan Feng[1], Joey Tianyi Zhou[2], Yatao Bian[3],**
**Qinghua Hu[1], Changqing Zhang[1]†**
College of Intelligence and Computing, Tianjin University[1]
A*STAR[2], Tencent AI Lab[3]

## Abstract

Out-of-distribution (OOD) detection is essential for model trustworthiness which aims to sensitively identify semantic OOD samples and robustly generalize for covariate-shifted OOD samples. However, we discover that the superior OOD detection performance of state-of-the-art methods is achieved by secretly sacrificing the OOD generalization ability. Specifically, the classification accuracy of these models could deteriorate dramatically when they encounter even minor noise. This phenomenon contradicts the goal of model trustworthiness and severely restricts their applicability in real-world scenarios. What is the hidden reason behind such a limitation? In this work, we theoretically demystify the "*sensitive-robust*" dilemma that lies in many existing OOD detection methods. Consequently, a theory-inspired algorithm is induced to overcome such a dilemma. By decoupling the uncertainty learning objective from a Bayesian perspective, the conflict between OOD detection and OOD generalization is naturally harmonized and a dual-optimal performance could be expected. Empirical studies show that our method achieves superior performance on standard benchmarks. To our best knowledge, this work is the first principled OOD detection method that achieves state-of-the-art OOD detection performance without compromising OOD generalization ability. Our code is available at https://github.com/QingyangZhang/DUL.

## 1 Introduction

Endowing machine learning models with out-of-distribution (OOD) detection and OOD generalization ability are both essential for their deployment in the open world [1, 2, 3]. We borrow an example of autonomous driving from [4] to demonstrate the motivation of these two tasks. Given a machine learning model trained on in-distribution (ID) data (top image in Fig. 1 (a)), OOD detection aims to sensitively perceive uncertainty arising upon outliers that do not belong to any known classes of training data [5] (bottom right image in Fig. 1 (a)). While OOD generalization expects machine learning models to be robust in the presence of unexpected noise or corruption, e.g., rainy or snowy weather (bottom left image in Fig. 1 (a)). In this paper, we reveal that many previous methods pursue OOD detection performance at a secret cost of sacrificing OOD generalization ability. To make things worse, we observe that some SOTA OOD detection methods may result in a catastrophic collapse in classification performance ($\sim$15% accuracy degradation) when encountering even slight noise. One pioneering work [4] makes a trade-off between OOD detection and OOD generalization, but the relationship between these two tasks is still largely unexplored. The learning objectives of these two tasks are seemingly conflicting at first glance. OOD detection encourages sensitive uncertainty awareness (highly uncertain prediction) on unseen data, while generalization expects the prediction to be confident and robust under unforeseeable distributional shifts. Previous work in OOD detection

---

*Work done during an internship at Tencent AI Lab.
†Correspondence to Changqing Zhang <zhangchangqing@tju.edu.cn>

38th Conference on Neural Information Processing Systems (NeurIPS 2024).

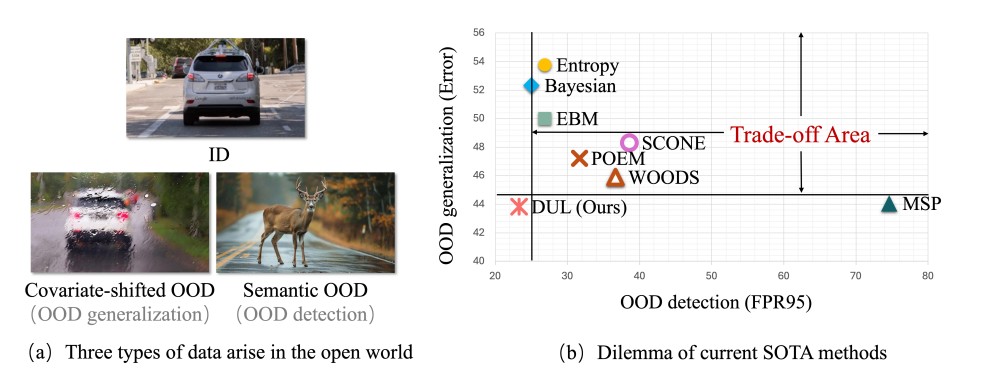

(a) Three types of data arise in the open world      (b) Dilemma of current SOTA methods

Figure 1: **(a)**: Models trained on in-distribution (ID) data inevitably encounter distributional shifts during their deployment. OOD generalization expects the model to correctly classify covariate-shifted data that undergoes noise or corruption due to environmental issues. OOD detection aims to identify samples that do not belong to any known classes for trustworthiness consideration. **(b)**: Limitations of current advanced OOD detection methods. We consider 8 representative OOD detection methods including the baseline method MSP [6] (without any OOD detection regularization), Entropy [7], EBM [8], Bayesian [9], SOTA OOD detection methods WOODS [10], POEM [11], recent advanced SCONE [4] which aims to seek for a good trade-off and the proposed DUL. All these methods exhibit a degraded generalization ability compared to baseline method MSP and lie in a trade-off area except our DUL. The goal of this paper is to understand and mitigate this phenomenon.

research area [4] characterizes the relationship between OOD detection and OOD generalization as a trade-off and thus striking for a balanced performance. However, this trade-off significantly limits the employment of current state-of-the-art OOD detection methods. Naturally, one might require the model to be aware of the OOD input for ensuring safety, but certainly does not expect to sacrifice the generalization ability, not to mention that the catastrophically collapsed classification performance under noise or corruption.

In this work, we first uncover the potential reason behind this limitation by characterizing the generalization error lower bound of previous OOD detection methods, which is referred to *sensitive-robust dilemma*. To overcome the dilemma, we devise a novel Decoupled Uncertainty Learning (DUL) framework for dual-optimal performance. The decoupled uncertainties are separately responsible for characterizing semantic OOD (detection) and covariate-shifted OOD (generalization). Thanks to the decoupled uncertainty learning objective, dual-optimal OOD detection and OOD generalization performance could be expected. Our emphasis lies on a particular category of OOD detection methods in the classification task, including max softmax probability (MSP) based model [6], energy-based model (EBM) [8] and Bayesian methods [9]. This selection offers two-fold advantages. First, MSP, EBM and Bayesian detectors encompass major OOD detection advances in classification task [5]. Second, numerous OOD detection works in diverse learning tasks (classification, object detection [12], time-series prediction and image segmentation) are all roughly related to classification [13]. The contributions of this paper are summarized as follows:

- This paper reveals that existing SOTA OOD detection methods may suffer from catastrophic degradation in terms of OOD generalization. That is, their superior OOD detection ability is achieved by (secretly) sacrificing OOD generalization ability. We theoretically demystify the sensitive-robust dilemma in learning objectives as the main reason behind such a limitation.

- In contrast to previous works that characterize OOD detection and generalization as conflictive learning tasks and thus implying an inevitable trade-off, we propose a novel learning framework termed Decoupled Uncertainty Learning (DUL) to successfully break through the limitation beyond a simple trade-off. Our DUL substantially harmonizes the conflict between OOD detection and OOD generalization, which achieves the best OOD detection performance without sacrificing the OOD generalization ability.

- We conduct extensive experiments on standard benchmarks to validate our findings. Our DUL achieves dual-optimal OOD detection and OOD generalization performance. To our best knowledge, DUL is the first method that gains state-of-the-art OOD detection performance without sacrificing OOD generalization ability.

## 2 Related works

**OOD detection** aims to indicate whether the input arises from unknown classes that are not present in training data, which is essential for model trustworthiness. In the classification task, the majority of advanced OOD detection methods include MSP detectors which characterize samples with lower max softmax probability as OOD [6, 14, 15, 7, 16]. EBM detectors identify high energy samples as OOD and frequently establish better performance than MSP detectors [8, 11, 17, 4], and various other types OOD detection methods such as distance-based detectors [18], non-parametric KNN-based detectors [19] which also show promises. According to the training paradigm, OOD detection methods can be split into auxiliary OOD-free and auxiliary OOD-required methods. Auxiliary OOD-free methods directly use the model pre-trained on ID data only for OOD detection. Another line of methods assumes that some OOD data is accessible during training and incorporates auxiliary outlier datasets (collected from websites or other datasets) for further enhancing OOD detection performance. By exposing the model to some semantic OOD during training, auxiliary OOD-required methods frequently outperform auxiliary OOD-free methods on commonly-used benchmarks [20, 5].

**OOD generalization** expects the model to be robust under unforeseeable noise or corruption [21, 22, 23, 24, 25]. Basically, OOD generalization expects invariant and confident prediction on OOD data. Examples include classic domain adaption (DA) methods which encourage the model's behavior to be invariant across different distributions [21, 26, 27]. Besides, test-time adaption (TTA) directly encourages confident predictions on OOD data by minimizing predictive entropy [28, 29, 30]. However, as we will show later, confident prediction and invariance are seemingly conflictive to OOD detection purpose and further imply an unavoidable trade-off. The most related work to our paper is SCONE [4], which strikes to keep a balance between OOD detection and generalization performance. We argue that such a trade-off is not necessary and the conflict can be elegantly eliminated.

**Uncertainty estimation in Bayesian framework.** In the Bayesian framework, predictive uncertainty can be regarded as an indicator of whether the input sample is prone to be OOD. Since OOD samples are unseen during training and thus should be of higher uncertainty than ID. The overall predictive uncertainty of a classification model can be decomposed into three factors according to their source, including data (aleatoric) uncertainty (AU), distributional uncertainty (DU), and model (epistemic) uncertainty (EU) [31, 9]. AU measures the natural complexity of the data (e.g., class overlap, label noise) and EU results from the difficulty of estimating the model parameters with finite training data. DU arises due to the mismatch between the distributions of test and training data. A line of classic measurement can be used to capture various types of uncertainty including entropy, mutual information, and differential entropy [9].

## 3 Preliminaries

We consider $K$-class classification task with classifier $f : \mathcal{X} \rightarrow \mathbb{R}^K$ parameterized by $\theta$, where $\mathcal{X}$ is the input space and $\mathcal{Y} = \{1, 2, ..., K\}$ denotes the target space. The model output $f_\theta(x)$ is considered as logits. The $k$-th element in logits is denoted as $f_k(x)$ indicates the confidence of predicting $x$ to class $k$. The predicted distribution $F(x)$ is obtained by normalizing $f(x)$ with the softmax function. We first formalize all possible distributions that the model might encounter.

- In-distribution $P_{\mathcal{X}\mathcal{Y}}^{\mathrm{ID}}$ which denotes the distribution of labeled training data.

- Covariate-shifted OOD $P_{\mathcal{X}\mathcal{Y}}^{\mathrm{COV}}$ which is relevant to OOD generalization. $P_{\mathcal{X}\mathcal{Y}}^{\mathrm{COV}}$ is of the same label space with ID. However, its marginal distribution $P_{\mathcal{X}}^{\mathrm{COV}}$ encounters shifts due to unexpected noise or corruption.

- Semantic OOD $P_{\mathcal{X}\mathcal{Y}'}^{\mathrm{SEM}}$ is the distribution of data that do not belong to any known class. Its label space has no overlap with the known ID label space, i.e., $\mathcal{Y}' \cap \mathcal{Y} = \emptyset$.

In the following paper, we omit the subscript for simplicity. The goal of OOD detection is to build a detector $G : \mathcal{X} \rightarrow [\mathrm{IN}, \mathrm{OUT}]$ to decide whether an input $x$ is semantic OOD data or not through a thresholding function $G$ deduced from classifier $f$

$$G_\gamma(x) = \begin{cases} \mathrm{IN} & g_f(x) \leq \gamma \\ \mathrm{OUT} & g_f(x) > \gamma \end{cases}, \tag{1}$$

where $\gamma$ is the threshold. $g_f$ is an OOD scoring function deduced from $f$, which is expected to assign a higher value to OOD than ID. For example, in MSP detectors, $g_f(x) = -\max_k F(x)$ where $F(x)$ is the predicted softmax probability (negative max softmax probability). In EBM detectors, $g_f$ is realized by the energy function $E(x; f) := -\log \sum_{i=1}^{K} e^{f_k(x)}$ and the semantic input of OOD should be of high energy [8]. Since it is difficult to foresee $P^{\text{SEM}}$ one will encounter, a board line of OOD detection works [7, 8, 9, 11, 17, 12, 32, 33] regularize the model on some auxiliary OOD data $P^{\text{SEM}}_{\text{train}}$ during training (e.g., data from the web or other datasets), and expect the model can learn useful heuristic to handle unknown test-time OOD $P^{\text{SEM}}_{\text{test}}$. The learning objective is shown as follows

$$\min_{\theta} \mathbb{E}_{(x,y)\sim P^{\text{ID}}}[\mathcal{L}_{\text{CE}}(f(x), y)] + \lambda \mathbb{E}_{\tilde{x}\sim P^{\text{SEM}}_{\text{train}}}[\mathcal{L}_{\text{reg}} f(\tilde{x})], \tag{2}$$

where $\mathcal{L}_{\text{CE}}$ is the standard cross entropy loss for the original classification task. $\mathcal{L}_{\text{reg}}$ is the OOD detection regularization term depending on the detector used, which generally encourages a high uncertainty on $P^{\text{SEM}}_{\text{train}}$. For example, $\mathcal{L}_{\text{reg}}$ is set to cross entropy between $F(x)$ and the uniform distribution for MSP detector [7]. In EBM detectors [8], $\mathcal{L}_{\text{reg}}$ is realized as a margin ranking loss to explicitly encourage a large energy gap between ID and semantic OOD. In this paper, we are interested in this setting for the following reasons:

1) In contrast to labeled data in supervised learning literature, auxiliary OOD data can be unlabeled and easy-to-collected in practice [11].

2) Most SOTA methods involve auxiliary outliers [5, 20] for superior performance.

3) Even under some strict assumptions that $P^{\text{SEM}}_{\text{train}}$ is unavailable, recent works utilize GAN [15], diffusion model [34] or sampling strategy [12] to generate "virtual" outliers for training.

Thus we believe this setting is promising and the cost of auxiliary outliers is minor given the importance of ensuring model trustworthiness. At test-time, the model is evaluated in terms of

- ID accuracy (ID-Acc $\uparrow$) which measures the model's performance on $P^{\text{ID}}$,

- OOD accuracy (OOD-Acc $\uparrow$) measures the OOD generalization ability on $P^{\text{COV}}$,

- False positive rate at 95% true positive rate (FPR95$\downarrow$) := $\mathbb{E}_{x\sim P^{\text{SEM}}_{\text{test}}}(\mathbb{I}(G_{\gamma}(x) = \text{IN}))$ measures the OOD detection ability, where $\gamma$ is chosen when true positive rate (TPR) is 95%. $\mathbb{I}$ is the indicator function. In OOD detection, ID samples are considered as negative.

It is worth noting that in the standard OOD detection setting [11, 4], the test OOD data should not have any overlapped classes or samples with training-time auxiliary OOD data $P^{\text{SEM}}_{\text{train}}$. Let $\mathcal{Y}^{\text{SEM}}_{\text{test}}$ and $\mathcal{Y}^{\text{SEM}}_{\text{train}}$ be the label space of $P^{\text{SEM}}_{\text{test}}$ and $P^{\text{SEM}}_{\text{train}}$ respectively, we have $\mathcal{Y}^{\text{SEM}}_{\text{test}} \cap \mathcal{Y}^{\text{SEM}}_{\text{train}} = \emptyset$. Otherwise, OOD detection would be a trivial problem.

## 4 Sensitive-robust Dilemma of Out-of-distribution Detection

In this section, we detail the limitation of current OOD detection methods: their OOD detection performance is achieved at the cost of generalization ability. This limitation implies the potential risk of SOTA OOD detection methods and underscores the urgent need for a better solution. Firstly, we re-examine representative OOD detection methods of six different types, including 1) baseline model MSP that is trained without any OOD detection regularization [6], 2) entropy-regularization (Entropy) that encourages high predictive entropy on OOD [7], which is devised for MSP detectors, 3) energy-regularization for EBM detectors that enforces the output with high energy score for OOD input [8], 4) Bayesian uncertainty learning that encourages high overall uncertainty on OOD [9], 5) state-of-the-art OOD detection methods WOODS [10] and POEM [11] 6) the most related SCONE [4] that seeks for a trade-off between OOD detection and generalization performance.

**Limitation of current OOD detection methods.** In Fig. 1 (b), we investigate current OOD detection methods in terms of OOD classification error and FPR95. The expected classifier should yield both low OOD classification error and FPR95. As it is observed, despite the superior OOD detection performance, all above methods significantly underperform the baseline MSP in terms of OOD generalization. By contrast, our method (DUL) successfully overcomes the limitation.

**Theoretical justification.** Toward understanding the limitation, we provide theoretical analysis for two types of most popular OOD detection methods, i.e., MSP and EBM detectors. Our analysis

identifies the "*sensitive-robust*" dilemma as the main reason behind such a limitation. The roadmap of our analysis is: (1) inspired by transfer learning theory, we first reveal that OOD detection regularization applied on semantic OOD may also affect the behavior of model on covariate-shifted OOD; (2) then we demonstrate why MSP detectors suffer from poor generalization by characterizing its generalization error bound; (3) we further identify that EBM methods [8] suffer from a similar drawback when incorporating with gradient-based optimization. First of all, we recap the definition of disparity discrepancy in transfer learning theory [35, 27].

**Definition 1** (Disparity with Total Variation Distance). *Given two hypotheses $f', f \in \mathcal{F}$ and distribution $P$, we define the Disparity with Total Variation Distance between them as*

$$\mathrm{disp}_P(f', f) = \mathbb{E}_P[TV(F_f || F_{f'})], \tag{3}$$

*where $F_{f'}, F_f$ are the class distributions predicted by $f', f$ respectively. $TV(\cdot||\cdot)$ is the total variation distance, i.e., $TV(F_f || F_{f'}) = \frac{1}{2} \sum_{k=1}^{K} ||F_{f,k} - F_{f',k}||$.*

**Definition 2** (Disparity Discrepancy with Total Variation Distance, DD with TVD). *Given a hypothesis space $\mathcal{F}$ and two distributions $P, Q$, the Disparity Discrepancy with Total Variation Distance (DD with TVD) is defined as*

$$d_{\mathcal{F}}(P, Q) := \sup_{f', f \in \mathcal{F}} (\mathrm{disp}_P(f', f) - \mathrm{disp}_Q(f', f)). \tag{4}$$

Disparity discrepancy (DD) measures the "distance" between two distributions $P, Q$ which considers the hypothesis space. DD is one of the most fundamental conceptions in transfer learning theory which constrains the behavior of hypothesis in $\mathcal{F}$ should not be dissimilar substantially on different distributions $P$ and $Q$. [3] If the DD between semantic OOD and covariate-shifted OOD is limited, one can suppose that OOD detection regularization applied to semantic OOD samples will also influence the model's behavior on covariate-shifted OOD. Thus encouraging high uncertainty on semantic OOD may also result in highly uncertain prediction on covariate-shifted OOD, which is potentially harmful to generalization ability. We first formalize this intuition for MSP detectors.

**Theorem 1** (Sensitive-robust dilemma). *Let $\mathcal{P}^{\mathrm{COV}}$, $P_{\mathrm{test}}^{\mathrm{SEM}}$ be the covariate-shifted OOD and semantic OOD distribution. $\mathrm{GError}_{P^{\mathrm{COV}}}(f)$ denotes standard cross entropy loss taking expectation on $P^{\mathrm{COV}}$, i.e., generalization error. Then we have*

$$\underbrace{\mathrm{GError}_{P^{\mathrm{COV}}}(f)}_{\text{OOD generalization error}} \geq C - \sqrt{\frac{1}{8\kappa^2} \mathbb{E}_{P_{\mathrm{test}}^{\mathrm{SEM}}} [\underbrace{\mathcal{L}_{\mathrm{reg}}(f)}_{\text{OOD detection loss}} - \log K]^{\frac{1}{2}}} - \frac{1}{2\kappa} d_{\mathcal{F}}(\mathcal{P}^{\mathrm{COV}}, \mathcal{P}_{\mathrm{test}}^{\mathrm{SEM}}),$$

$$\tag{5}$$

*where $\mathcal{L}_{\mathrm{reg}}$ is the OOD detection loss devised for MSP detectors defined in [7], i.e., cross-entropy between predicted distribution $F(x)$ and uniform distribution. $d_{\mathcal{F}}(P^{\mathrm{COV}}, P_{\mathrm{test}}^{\mathrm{SEM}})$ is DD with TVD that measures the dissimilarity of covariate-shifted OOD and semantic OOD. $C$ and $\kappa$ are both some constants depending on hypothesis space $\mathcal{F}$, $P^{\mathrm{COV}}$ and $P_{\mathrm{test}}^{\mathrm{SEM}}$.*

The proof is deferred in Appendix A. Theorem 1 demonstrates that for MSP detectors, the OOD detection objective conflicts with OOD generalization. The model's generalization error lower bound is negatively correlated with OOD detection loss that the model tries to minimize. Thus given a limited $d_{\mathcal{F}}$, pursuing low OOD detection loss on $P_{\mathrm{test}}^{\mathrm{SEM}}$ will also inevitably result in highly uncertain prediction on $P^{\mathrm{COV}}$. It is worth noting that such an interpretative theorem is applicable for all MSP-based OOD detectors no matter whether the model involves $P_{\mathrm{train}}^{\mathrm{SEM}}$ during training or not. Since the inherent motivation of OOD detection methods lies in minimizing the OOD detection loss in $P_{\mathrm{test}}^{\mathrm{SEM}}$, regardless of the training strategies used.

**Why a limited $d_{\mathcal{F}}(P^{\mathrm{COV}}, P_{\mathrm{test}}^{\mathrm{SEM}})$ is practical?** In Theorem 1, $d_{\mathcal{F}}(P^{\mathrm{COV}}, P_{\mathrm{test}}^{\mathrm{SEM}})$ measures the dissimilarity between $P^{\mathrm{COV}}$ and $P_{\mathrm{test}}^{\mathrm{SEM}}$. It seems that this lower bound will be very small and trivial when $d_{\mathcal{F}}(P^{\mathrm{COV}}, P_{\mathrm{test}}^{\mathrm{SEM}})$ is large enough. However, since the semantic OOD samples can be any samples that do not belong to ID classes, one can suppose that semantic OOD samples are extremely diverse and some are of high similarity with ID and covariate-shifted OOD [37]. Detecting these "ID-like" OOD samples is inherently the core challenge of OOD detection [11, 17, 38]. Thus, it is reasonable to assume a limited $d_{\mathcal{F}}(P^{\mathrm{COV}}, P_{\mathrm{test}}^{\mathrm{SEM}})$. We provide more discussions in the Appendix D.2.

---

[3] Encompassed by [36] as a special case, our definition is realized using TVD for theoretical convenience.

As presented before, the key limitation of MSP detectors is that they enforce high-entropy prediction on semantic OOD. We proceed to reveal that EBM detectors suffer from similar issues due to the natural property of gradient-based optimization. For EBM detectors, $\mathcal{L}_{\mathrm{reg}}$ is defined by [8] is

$$\mathcal{L}_{\mathrm{reg}} = \mathbb{E}_{\tilde{x} \sim P_{\mathrm{train}}^{\mathrm{SEM}}}[\max(m_{\mathrm{OUT}} - E(\tilde{x}), 0)]^2 + \mathbb{E}_{x \sim P^{\mathrm{ID}}}[\max(0, E(x) - m_{\mathrm{IN}})]^2, \qquad (6)$$

which constrains the energy score $E(x; f) := -log \sum_{k=1}^{K} e^{f_k(x)}$ of ID sample $x$ to be lower than that of OOD sample $\tilde{x}$. $m_{\mathrm{IN}}, m_{\mathrm{OUT}}$ are manually selected margins. Although such regularization does not indicate high entropy prediction at first glance, unfortunately, we demonstrate that EBM detectors also tend to uncertain prediction when equipped with gradient-based optimization. Here we focus on Gradient Descent (GD) as a showcase. In each training epoch $t$, model $f_\theta$ is updated with GD as $\theta_{t+1} = \theta_t - \eta \nabla_\theta \mathcal{L}_{\mathrm{reg}}$, where $\eta$ is the learning rate. For any sample $\tilde{x}$ drawn from $\mathcal{P}^{\mathrm{SEM}}$, the gradient $\nabla_\theta \mathcal{L}_{\mathrm{reg}}$ can be written as

$$2 \sum_{k=1}^{K} \nabla f_k(\tilde{x}) \underbrace{[e^{f_k(\tilde{x})}(\sum_{k=1}^{K} e^{f_k(\tilde{x})})^{-1}]}_{\text{higher for larger } f_k(\tilde{x})}(m_{\mathrm{OUT}} - E(\tilde{x})), \qquad (7)$$

where $m_{\mathrm{OUT}} - E(\tilde{x}) > 0$ (otherwise the gradient is zero) and $f_k(\tilde{x})$ is the predicted logits on the $k$-th class. For sample $\tilde{x}$, when class $k$ has larger predicted logit, it contributes more to the overall gradient $\nabla_\theta \mathcal{L}_{\mathrm{reg}}$ and thus could obtain more optimization efforts during backpropagation. Eventually, one can infer that when an EBM detector is about to converge, it tends to high-entropy prediction on $P_{\mathrm{train}}^{\mathrm{SEM}}$ accordingly. Incorporating this into the established Theorem 1, this is likely to harm the generalization ability of $P^{\mathrm{COV}}$. Empirical evidence can support this supposition (see Table 14 in Appendix C). Therefore both MSP and EBM detectors face the "*sensitive-robust*" dilemma.

## 5  Decoupled Uncertainty Learning

We demonstrate how to handle the dilemma between OOD detection and generalization by decoupled uncertainty learning in the Bayesian framework. Unlike the most related work [4] which aims to seek a good trade-off, our method successfully gets out of the aforementioned sensitive-robust dilemma.

**Uncertainty Estimation in Bayesian Framework.** We first revisit the theoretical properties of different types of uncertainty in a Bayesian framework. Non-Bayesian classifiers consider the model's output $f(x)$ as logits, which is then normalized with softmax to directly model predictive categoricals $p(\hat{y}|x)$. While in Bayesian framework [9], $f(x)$ is considered as parameters of a Dirichlet distribution $p(\mu|x)$ firstly, which is used to model the prior of predictive categoricals $p(\hat{y}|x)$ by

$$p(\mu|x) = \mathrm{Dir}(\mu|\boldsymbol{\alpha}) = \frac{\Gamma(\alpha_0)}{\prod_{k=1}^{K} \Gamma(\alpha_k)} \prod_{k=1}^{K} \mu_k^{\alpha_k - 1}, \ \boldsymbol{\alpha} = f(x), \qquad (8)$$

where $\mathrm{Dir}(\mu|\boldsymbol{\alpha})$ is Dirichlet distribution and $\boldsymbol{\alpha}$ is the concentration parameters of Dirichlet. The sum of all $\alpha_k \in \boldsymbol{\alpha}$ (noted as $\alpha_0$) is so called the strength of the Dirichlet distribution, i.e., $\alpha_k > 0, \alpha_0 = \sum_k \alpha_k$. After obtaining prior $p(\mu|x, \theta)$, the final predicted posterior $p(\hat{y}|x)$ over class labels is given by calculating the mean of the Dirichlet prior

$$\underbrace{p(\hat{y}|x, \theta)}_{\text{overall uncertainty}} = \int \overbrace{p(\hat{y}|\mu)}^{\text{data uncertainty}} \underbrace{p(\mu|x, \theta)}_{\text{distributional uncertainty}} \ d\mu. \qquad (9)$$

From a Bayesian perspective, given deterministic parameters $\theta$, the overall uncertainty of final prediction $p(\hat{y}|x)$ can be decomposed into two factors, including data (aleatoric) uncertainty (AU) and distributional uncertainty (DU). DU lies in $p(\mu|x, \theta)$ which is defined as uncertainty due to the mismatch between the distributions of test and train data. AU is described by $p(\hat{y}|\mu)$ which captures the natural complexity of the data (e.g., class-overlap) [9]. By definition, OOD detection is primarily associated with DU which is only a part of the overall uncertainty. While generalization is related to the overall uncertainty of $p(\hat{y}|x)$ as we mentioned in related works (both AU and DU). One essential property of DU is that it can be high even if the expected categorical $p(\hat{y}|x, \theta)$ expresses low overall

uncertainty. Such a property is well suited to achieve OOD detection and generalization jointly since high DU no longer necessarily indicates high overall uncertainty.

**Decoupled Uncertainty Learning.** While the aforementioned Bayesian framework enjoys theoretical potentiality, its learning object [9] lacks consideration of OOD generalization. Similar to other OOD detection methods, it also directly enforces high overall uncertainty on OOD

$$\min_{\theta} \mathbb{E}_{\mathcal{P}^{\text{ID}}} \text{KL}(p(y|x))||p(\hat{y}|x)) + \mathbb{E}_{\mathcal{P}^{\text{SEM}}_{\text{train}}} \text{KL}(p(\mu|\tilde{x}))||\text{Dir}(\mu|\alpha = \mathbf{1})), \tag{10}$$

where $p(y|x), p(\hat{y}|x)$ are the ground-truth distribution and predicted distribution on ID. The model's prediction on OOD is enforced to be close to a rather flat Dirichlet distribution. It is worth noting that $\text{Dir}(\mu|\alpha = \mathbf{1})$ means all classes are equiprobable, and the entropy of the final prediction is maximized. As shown in Fig. 1 (b), the vanilla Bayesian method [9] also suffers from degraded OOD generalization performance. To this end, we propose **D**ecoupled **U**ncertainty **L**earning (DUL), a novel OOD detection regularization method that explicitly encourages high DU on OOD samples without affecting the overall uncertainty. Similarly to previous OOD detection methods [8], our DUL is also devised in a finetune manner for effectiveness. Given a classifier $f_{\theta_0}$ well pre-trained on $P^{\text{ID}}$, the goal of DUL lies in enhancing its OOD detection performance without sacrificing any generalization ability. Specifically, we finetune the model by encouraging higher DU but non-increased overall uncertainty on $P^{\text{SEM}}_{\text{train}}$. The learning objective of DUL is

$$\min_{\theta} \underbrace{\mathbb{E}_{(x,y)\sim P^{\text{ID}}}[\mathcal{L}_{\text{CE}}(f(x), y)]}_{\text{ID classification}} + \lambda \underbrace{\mathbb{E}_{\tilde{x}\sim P^{\text{SEM}}_{\text{train}}}||\max(0, (h_0 + m_{\text{OUT}}) - h)||_{\tau}}_{\text{high distributional uncertainty (detection)}}$$

$$\text{s.t.} \quad \underbrace{H(p(\hat{y}|\tilde{x})) = H(p_0(\hat{y}|\tilde{x}))}_{\text{non-increased overall uncertainty (generalization)}} \quad \forall \tilde{x} \sim P^{\text{SEM}}_{\text{train}}, \tag{11}$$

where $H(\cdot)$ is the entropy. $p(\hat{y}|\tilde{x})$ and $p_0(\hat{y}|\tilde{x})$ are the predicted distribution on semantic OOD data $\tilde{x}$ after and before finetuning. The first term is the original ID classification loss. The second term is OOD detection loss, which encourages high DU on outlier $\tilde{x}$. $m_{\text{OUT}}$ and $\tau > 0$ are hyperparameters. $h_0, h$ are DU on $\tilde{x}$ before and after finetuning. Here we measure DU with the differential entropy $(h[p(\mu|\tilde{x}, \theta)] = -\int_{S^{K-1}} p(\mu|\tilde{x}) \ln(p(\mu|\tilde{x})) d\mu$, $S$ is a $K$-simplex). We refer interested readers to the Appendix D.1 for mathematical details. The third term constraining on $H(p(\hat{y}|\tilde{x}))$ avoids increment of overall uncertainty during finetuning and thus the generalization ability can be retained. Considering the difficulty of constrained optimization, we convert Eq. 11 into an unconstrained form and get our final minimizing objective

$$\underbrace{\mathbb{E}_{P^{\text{ID}}}[\mathcal{L}_{\text{CE}}(f(x), y)]}_{\text{ID classification}} + \mathbb{E}_{P^{\text{SEM}}_{\text{train}}} \{\lambda \underbrace{||\max(0, (h_0 + m_{\text{OUT}}) - h)||_{\tau}}_{\text{high distributional uncertainty}} + \underbrace{\gamma \text{KL}(p(\hat{y}|\tilde{x})||p_0(\hat{y}|\tilde{x}))}_{\text{unchanged overall uncertainty}} \},$$

$$\tag{12}$$

where $\gamma$ is hyperparameter. In contrast to previous Bayesian method [9], DUL only encourages high DU rather than overall uncertainty on OOD and explicitly discourages high entropy in the final prediction. The implementation details are in Appendix D.1.

## 6 Experiment

We conduct experiments to validate our analysis and the superiority of DUL. The questions to be verified are Q1 Motivation. To what extent does OOD detection conflict with OOD generalization in previous methods? Q2 Effectiveness. Does DUL achieve better OOD detection and generalization performance compared to its counterparts? Q3 Interpretability. Does the proposed method well decouple uncertainty as expected?

### 6.1 Experimental Setup

Our settings follow the common practice [8, 11, 20, 5] in OOD detection. Here we present a brief description and more details about datasets, metrics, and implementation are in Appendix B.1 and B.2.

**Datasets.** ∘ **ID datasets** $P^{\text{ID}}$. We train the model on different ID datasets including CIFAR-10, CIFAR-100 and ImageNet-200 (a subset of ImageNet-1K [39] with 200 classes). ∘ **Auxiliary OOD datasets** $P^{\text{SEM}}_{\text{train}}$. In CIFAR experiments, we use ImageNet-RC as $P^{\text{SEM}}_{\text{train}}$. ImageNet-RC is a down-sampled variant of the original ImageNet-1K which is widely adopted in previous OOD detection

Table 1: OOD detection and generalization performance comparison. Substantial ($\geq 0.5$) improvement and degradation compared to the baseline MSP [6] (training without any OOD detection regularization) are highlighted in blue or red, respectively. The **best** and second-best results are in bold or underlined. DUL is the only method that achieves SOTA OOD detection performance (mostly the best or second best) without sacrificing generalization i.e., the value of the entire row is either blue or black. Full results with standard deviation and diverse types of corruption are in Appendix C.

| $\mathcal{P}^{ID}/\mathcal{P}^{SEM}_{train}$ | Method | Model generalization | | OOD detection | | |
|---|---|---|---|---|---|---|
| | | ID-Acc ↑ | OOD-Acc ↑ | FPR ↓ | AUROC ↑ | AUPR ↑ |
| CIFAR-10 / None | MSP | 96.11 | 87.35 | 41.96 | 89.28 | 68.00 |
| | EBM (pretrain) | 96.11 | 87.35 | 32.45 | 89.34 | 75.22 |
| | Maxlogits | 96.11 | 87.35 | 32.90 | 89.26 | 74.47 |
| | Mahalanobis | 96.11 | 87.35 | 32.53 | 93.93 | 74.96 |
| CIFAR-10 / ImageNet-RC | Entropy | 96.04 | 72.57 | 6.63 | 98.72 | 94.00 |
| | EBM (finetune) | **96.10** | 79.03 | 3.61 | 98.39 | 94.88 |
| | POEM | 94.32 | 78.89 | **3.32** | **98.99** | **99.38** |
| | DPN | 95.69 | 85.52 | 4.28 | 98.53 | 94.93 |
| | WOODS | 96.01 | 80.14 | 7.12 | 98.45 | 92.46 |
| | SCONE | 95.96 | 78.80 | 7.02 | 98.45 | 92.46 |
| | DUL (ours) | 96.02±0.00 | **88.01±0.29** | 5.89±0.12 | 98.47±0.02 | 92.44±1.29 |
| | DUL† (ours) | 96.04±0.00 | 87.53±0.49 | 5.99±0.06 | 98.28±0.01 | 98.40±0.13 |
| CIFAR-10 / TIN-597 | Entropy | 95.94 | 80.51 | 11.60 | 97.93 | 92.16 |
| | EBM (finetune) | 95.38 | 83.67 | 19.36 | 87.51 | 83.63 |
| | POEM | 95.44 | 83.17 | 24.34 | 86.83 | 94.25 |
| | DPN | 94.39 | 79.23 | 17.27 | 94.92 | 87.67 |
| | WOODS | 95.57 | 84.68 | 7.58 | **98.29** | 93.08 |
| | SCONE | 95.19 | 84.68 | 8.02 | 98.21 | 93.08 |
| | DUL (ours) | **96.06±0.01** | 87.93±0.39 | **6.87±0.67** | 98.21±0.01 | 91.29±1.39 |
| | DUL† (ours) | 95.94±0.01 | **88.10±0.07** | 10.34±0.11 | 97.67±0.01 | **98.59±0.06** |
| CIFAR-100 / None | MSP | 80.99 | 55.95 | 74.63 | 80.19 | 42.59 |
| | EBM (pretrain) | 80.99 | 55.95 | 67.42 | 82.67 | 49.35 |
| | Maxlogits | 80.99 | 55.95 | 69.32 | 82.30 | 47.60 |
| | Mahalanobis | 80.99 | 55.95 | 61.51 | 85.97 | 56.10 |
| CIFAR-100 / ImageNet-RC | Entropy | 80.21 | 45.48 | 22.29 | 95.33 | 82.34 |
| | EBM (finetune) | 80.53 | 48.14 | 13.47 | 96.78 | 87.84 |
| | POEM | 78.15 | 42.18 | **9.89** | **97.79** | **98.40** |
| | DPN | 78.90 | 50.14 | 18.36 | 95.42 | 74.45 |
| | WOODS | 80.69 | 54.38 | 38.15 | 92.01 | 71.79 |
| | SCONE | 80.80 | **56.73** | 47.60 | 89.61 | 65.29 |
| | DUL (ours) | **81.30±0.04** | 56.27±3.29 | 12.49±0.05 | 95.24±0.01 | 86.72±0.58 |
| | DUL† (ours) | 81.23±0.05 | 55.41±0.54 | 11.12±0.62 | 95.46±0.36 | 96.49±0.13 |
| CIFAR-100 / TIN-597 | Entropy | 80.15 | 46.25 | 26.88 | 93.50 | 79.81 |
| | EBM (finetune) | 79.94 | 50.00 | 26.87 | 91.68 | 80.08 |
| | POEM | 78.68 | 52.53 | 32.71 | 91.30 | 94.65 |
| | DPN | 78.44 | 47.67 | 24.99 | 93.55 | 81.63 |
| | WOODS | 79.26 | 53.13 | 36.71 | 92.15 | 73.42 |
| | SCONE | 79.53 | 52.70 | 35.60 | 92.47 | 73.58 |
| | DUL (ours) | **80.85±0.06** | 56.19±2.33 | 23.32±1.22 | **94.48±0.12** | 80.82±2.63 |
| | DUL† (ours) | 80.50±0.06 | **56.22±1.66** | 22.75±0.78 | 90.88±0.08 | **96.33** |
| ImageNet-200 / None | MSP | 85.15 | 74.84 | 58.23 | 86.98 | 82.27 |
| | EBM (pretrain) | 85.15 | 74.84 | 51.94 | 88.18 | 84.75 |
| | Maxlogits | 85.15 | 74.84 | 51.62 | 88.30 | 84.71 |
| ImageNet-200 / ImageNet-800 | Entropy | 84.92 | 74.75 | 53.62 | 89.05 | 85.02 |
| | EBM (finetune) | 84.14 | 73.31 | 59.73 | 87.54 | 82.81 |
| | DPN | 84.87 | 74.40 | 63.84 | 87.18 | 80.69 |
| | WOODS | 84.99 | 74.98 | 51.71 | 88.30 | 84.80 |
| | SCONE | 84.93 | 74.91 | 52.52 | 88.19 | 84.50 |
| | DUL (ours) | **85.65±0.07** | **75.59±0.12** | **49.14±0.13** | **89.27±0.03** | **85.62±0.03** |

works [8, 11, 17]. We also conduct experiments on the recent TIN-597 [20] as an alternative. When ImageNet-200 is ID, the remaining 800 classes termed ImageNet-800 are considered as $P^{SEM}_{train}$. ○ **OOD detection test sets** $P^{SEM}_{test}$ are a suite of diverse datasets introduced by commonly used benchmark [5]. In CIFAR experiments, we use SVHN [40], Places365 [41], Textures [42], LSUN-R, LSUN-C [43] and iSUN [44] as $P^{SEM}_{test}$. When $P^{ID}$ is ImageNet-200, $P^{SEM}_{test}$ consists of iNaturlist [45], Open-Image [46], NINCO [47] and SSB-Hard [48]. It is worth noting that in standard OOD detection settings, there should be no overlapped classes between $P^{ID}$, $P^{SEM}_{train}$ and $P^{SEM}_{test}$, otherwise OOD detection is a trivial problem. ○ **OOD generalization test sets** $P^{COV}$ is the original ID test set corrupted with additive Gaussian noise of $\mathcal{N}(0, 5)$, following [4]. Besides, we also conduct experiments on CIFAR10-C, CIFAR100-C and ImageNet-C which involve 15 diverse types of different noise or corruption (e.g., snow, rain, frost, fog...) in Appendix C.

**Metrics.** For OOD detection performance evaluation, we report the average FPR95, AUROC and AUPR to be consistent with [11]. OOD generalization ability is compared in terms of classification accuracy (OOD-Acc). Besides, we also report classification accuracy on ID test sets (ID-Acc).

**Compared methods.** We compare DUL with a board line of OOD detection methods, including auxiliary OOD required and auxiliary OOD free methods. ○ **Auxiliary OOD-free methods** do not require $P_{\text{train}}^{\text{SEM}}$ during training, including MSP [6], Maxlogits [49], pretrained EBM [8] and Mahalaobis [18]. ○ **Auxiliary OOD-required methods** explicitly regularize the model on $P_{\text{train}}^{\text{SEM}}$, including entropy-regularization (Entropy) [7], finetuned EBM [8], DPN of Bayesian framework [9], POEM [11] and WOODS [10]. We also compare our DUL to recent advanced SCONE [4] which aims to keep a balance between OOD detection and generalization.

## 6.2 Experimental Results

**Dilemma between OOD detection and generalization (Q1).** We validate the dilemma mentioned before in Fig. 1. As shown in Tab. 1, though many advanced methods establish superior OOD detection performance, their OOD generalization degrades a lot. For example, recent SOTA POEM achieves nearly perfect OOD detection performance on CIFAR10 when ImageNet-RC serves as $P_{\text{train}}^{\text{SEM}}$ with 3.32% false positive error rate (FPR95). However, its OOD-Acc drops a lot (about 10%) compared to baseline MSP. This phenomenon is also observed in other advanced methods. To further detail this phenomenon, we reduce the weight of OOD detection regularization terms in Entropy and finetuned EBM and show the performance on both OOD detection and generalization. As shown in Table 3, when the regularization strength increases, OOD detection performance improves (lower FPR.), while the OOD generalization performance degrades (higher error rate).

**OOD detection and generalization ability (Q2).** As shown in Tab. 1, DUL establishes strong overall performance in terms of both OOD detection and generalization. We highlight a few essential observations: 1) **Compared to auxiliary OOD free methods**, DUL establishes substantial improvement due to additional regularization on auxiliary outliers. 2) **Compared to auxiliary OOD required methods**, our method achieves superior OOD detection performance without sacrificing generalization ability. Meanwhile, previous OOD detection methods commonly exhibit severely degraded classification accuracy, with many cases increasing by more than 10% error rate. 3) **Comparison to the most related work SCONE [4].** Despite recent advanced SCONE simultaneously considering both two targets, we observe that it can be hard to find a good trade-off. In contrast, dual-optimal OOD detection and generalization performance is achieved by our DUL. Noted that DUL is the only method that achieves *state-of-the-art* detection performance (mostly the best or second best) without degraded generalization ability (no red values in the entire row). The sensitive-robust dilemma is no longer observed in our method. These observations justify our expectation of DUL. 4) **Combining with existing methods.** Besides, to further demonstrate the effectiveness of the proposed DUL, we also add the unchanged overall uncertainty term in Eq.12 to the original Entropy and finetuned EBM. The results in Table 2 show that DUL regularization can also benefit EBM. However, combining Entropy with our regularization can not improve the accuracy substantially. This is not surprising, since the target of Entropy (high entropy prediction) and our DUL (non-increased entropy) directly conflict according to Theorem 1. 5) **Comparison to methods with an extra OOD detect branch.** Different from aforementioned methods, a line of recent OOD detectors [50, 51, 52, 17] employ extra output branches aside from the classification logits (with a shared backbone for feature extraction). For these OOD detectors, our theoretical analysis is not directly applicable and further analysis from a feature learning perspective may be needed in future work. However, the proposed DUL is devised in a finetune manner. Compared to OOD detectors with extra output branches that requires re-training the classifier from scratch, DUL can be applied to any pre-trained model (e.g., from torchvision, huggingface), with modest computation overhead.

**Visualization of estimated uncertainty (Q3).** To evaluate the uncertainty estimation, we visualize the distribution of ID (CIFAR-10) and OOD (SVHN) samples in terms of uncertainty. As we can see in Fig. 2 (b), our DUL establishes a distinguishable (distributional) uncertainty gap between test-time ID and OOD data, which indicates a good sensitiveness for OOD detection. By contrast, the baseline method MSP (Fig.2 (a)) can not effectively discriminate ID and OOD. Besides, we visualize the predictive entropy (overall uncertainty) on covariate-shifted OOD (CIFAR-10 with Gaussian noise) in Fig. 2 (c), our DUL yields much lower entropy compared to other methods. Besides, we visualize the data uncertainty on semantic OOD test data (Textures) when CIFAR-10 is ID in Fig. 6.2. The

investigated methods are 1) pretrained model training on ID dataset only, 2) finetuned model with OOD detection regularization (ablating the last term in Eq.12), and 3) finetuned model with the full DUL method described by Eq.12. As shown in Fig. 6.2, to keep the overall uncertainty and enlarge the distributional uncertainty (for OOD detection), the data uncertainty must be reduced. We use Eq.17 from [9] to calculate data uncertainty. The distributional uncertainty is shifted by subtracting that on ID dataset. These results meet our expectation.

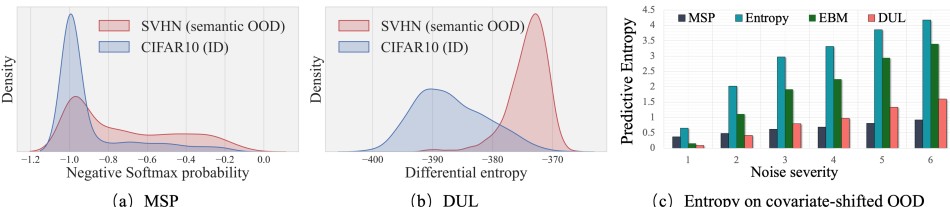

(a) MSP          (b) DUL          (c) Entropy on covariate-shifted OOD

Figure 2: Visualization of different types of uncertainty estimated by DUL.

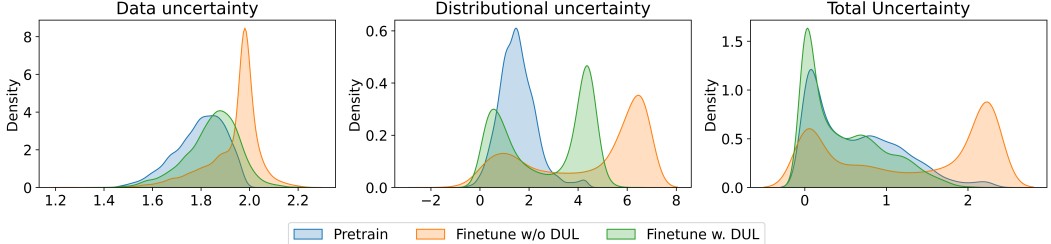

Figure 3: Visualization of uncertainty on semantic OOD test dataset when CIFAR-10 is ID dataset. Without DUL (orange), all three types of uncertainty will increase altogether. In contrast, DUL (green) increases the DU but decreases the AU, which further lead to unchanged overall uncertainty.

Table 2: Additional results when equip DUL to existing methods i.e., Entropy and finetuned EBM. ID dataset is CIFAR-10. $P_{\text{train}}^{\text{SEM}}$ is ImageNet-RC. $P_{\text{test}}^{\text{Cov}}$ is the original CIFAR-10 testset corrupted by Gaussian noise $\mathcal{N}(0, 5)$.

| Method | Model generalization | | OOD detection | |
|---|---|---|---|---|
| | ID-Acc ↑ | OOD-Acc ↑ | FPR ↓ | AUC ↑ |
| Entropy | 96.04 | 72.57 | 6.63 | 98.72 |
| EBM (finetune) | 96.10 | 79.03 | 3.61 | 98.39 |
| POEM | 94.32 | 78.89 | 3.32 | 98.99 |
| EBM w. DUL | 95.19 | 87.45 | 6.17 | 98.28 |
| Entropy w. DUL | 96.10 | 87.41 | 29.56 | 95.92 |
| DUL | 96.02 | 88.01 | 5.89 | 98.47 |
| DUL$^\dagger$ | 96.04 | 87.53 | 5.99 | 98.28 |

Table 3: We tune the weight of OOD detection regularization term for EBM as well as Entropy and report the FPR (OOD detection metric) and error rate (Err, OOD generalization metric). The experimental settings are the same with Table 2.

| | Entropy | | | EBM | |
|---|---|---|---|---|---|
| $\lambda$ | OOD-Err ↓ | FPR ↓ | $\lambda$ | OOD-Err ↓ | FPR ↓ |
| 0 | 9.55 | 35.15 | 0 | 9.55 | 20.57 |
| $5 \times 10^{-4}$ | 13.58 | 8.36 | $1 \times 10^{-4}$ | 9.46 | 14.69 |
| $5 \times 10^{-3}$ | 15.48 | 6.37 | $1 \times 10^{-3}$ | 10.32 | 13.54 |
| $5 \times 10^{-2}$ | 17.97 | 5.71 | $1 \times 10^{-2}$ | 16.43 | 8.15 |
| $5 \times 10^{-1}$ | 18.53 | 5.60 | $1 \times 10^{-1}$ | 24.38 | 6.11 |

# 7 Conclusion

This paper provides both theoretical and empirical analysis towards understanding the dilemma between OOD detection and generalization. We demonstrate that the superior OOD detection performance of current advances are achieved at the cost of generalization ability. The theory-inspired algorithm successfully removes the conflict between previous OOD detection and generalization methods. For SOTA OOD detection performance, our implementation assumes that auxiliary outliers are available during training. This limitation is noteworthy for our DUL as well as the most existing SOTA OOD detection methods. We argue that this added cost is minor and reasonable given the significance of ensuring model trustworthiness in open-environments. Reducing the dependency on auxiliary OOD data can be an interesting research direction for the future exploration.

# Acknowledgement

This work was supported by the National Natural Science Foundation of China (62376193 and 61925602). The authors thank NeurIPS anonymous peer reviewers for their helpful suggestions.

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

# Appendices

## A Proofs

First, we recap the definitions of Disparity with Total Variation Distance and Disparity Discrepancy.

**Definition 3** (Disparity with Total Variation Distance). *Given two hypotheses $f', f \in \mathcal{F}$ and distribution $P$, we define the Disparity with Total Variation Distance between them as*

$$\mathrm{disp}_P(f', f) = \mathbb{E}_P[TV(F_f||F_{f'})], \tag{13}$$

*where $F_f, F_{f'}$ are the class distributions predicted by $f', f$ respectively. $TV(\cdot||\cdot)$ is the total variation distance (TVD), i.e., $TV(F_f||F_{f'}) = \frac{1}{2}\sum_{k=1}^{K}||F_{f,k} - F_{f',k}||_1$.*

**Definition 4** (Disparity Discrepancy with Total Variation Distance, DD with TVD). *Given a hypothesis space $\mathcal{F}$ and two distributions $P, Q$, the Disparity Discrepancy with Total Variation Distance (DD with TVD) is defined as*

$$d_{\mathcal{F}}(P, Q) := \sup_{f', f \in \mathcal{F}} (\mathrm{disp}_P(f', f) - \mathrm{disp}_Q(f', f)). \tag{14}$$

Since TVD is a distance measurement of two distribution. It yields the triangle equality. That is, for any distribution $P_{\mathcal{X}}$ support on $\mathcal{X}$ and hypotheses $f^1, f^2$ and $f^3 \in \mathcal{F}$, we have

$$
\begin{aligned}
\mathrm{disp}_{P_{\mathcal{X}}}(f^1, f^2) &\leq \mathbb{E}_{x \sim P_{\mathcal{X}}}[TV(F_{f^1}(x)||F_{f^3}(x))] + \mathbb{E}_{x \sim P_{\mathcal{X}}}[TV(F_{f^2}(x)||F_{f^3}(x))], \\
\mathrm{disp}_{P_{\mathcal{X}}}(f^1, f^2) &\geq \mathbb{E}_{x \sim P_{\mathcal{X}}}[TV(F_{f^1}(x)||F_{f^3}(x))] - \mathbb{E}_{x \sim P_{\mathcal{X}}}[TV(F_{f^2}(x)||F_{f^3}(x))].
\end{aligned}
\tag{15}
$$

To prove Theorem 1, we need the following lemmas.

**Lemma 1.** *For any $f \in \mathcal{F}$, we have*

$$\mathbb{E}_{P^{\mathrm{COV}}} TV(F_f||U) \leq \mathbb{E}_{P_{\mathrm{test}}^{\mathrm{SEM}}} TV(F_f||U) + d_{\mathcal{F}}(P^{\mathrm{COV}}, P_{\mathrm{test}}^{\mathrm{SEM}}) + \lambda \tag{16}$$

*where $\lambda$ is a constant independent of $f$. $U$ is the $K$-classes uniform distribution. $P^{\mathrm{COV}}$ is the covariate-shifted OOD distribution. $P_{\mathrm{test}}^{\mathrm{SEM}}$ is the semantic OOD distribution.*

*Proof.* Let $f^*$ be the hypothesis which jointly minimizes the total variance distance between the predicted distribution $F_f$ with uniform distribution $U$ taking expectation on $P^{\text{COV}}$ and $P_{\text{test}}^{\text{SEM}}$, which is to say

$$f^* := \underset{f \in \mathcal{F}}{\arg\min}\{\mathbb{E}_{x \sim P^{\text{COV}}}[TV(F_f(x)||U)] + \mathbb{E}_{x \sim P_{\text{test}}^{\text{SEM}}}[TV(F_f(x)||U)]\}. \tag{17}$$

Set $\lambda = \mathbb{E}_{x \sim P^{\text{COV}}}[TV(F_{f^*}(x)||U)] + \mathbb{E}_{x \sim P_{\text{test}}^{\text{SEM}}}[TV(F_{f^*}(x)||U)]$, then by the triangle equality we have

$$
\begin{aligned}
\mathbb{E}_{P^{\text{COV}}}TV(F_f||U) &\leq \text{disp}_{P^{\text{COV}}}(f, f^*) + \mathbb{E}_{P^{\text{COV}}}TV(F_{f^*}||U) \\
&\leq \mathbb{E}_{P_{\text{test}}^{\text{SEM}}}TV(F_f||U) - \mathbb{E}_{P_{\text{test}}^{\text{SEM}}}TV(F_f||U) + \text{disp}_{P^{\text{COV}}}(f, f^*) + \mathbb{E}_{P^{\text{COV}}}TV(F_{f^*}||U) \\
&\leq \mathbb{E}_{P_{\text{test}}^{\text{SEM}}}TV(F_f||U) + \mathbb{E}_{P_{\text{test}}^{\text{SEM}}}TV(F_{f^*}||U) \\
&\quad - \text{disp}_{P_{\text{test}}^{\text{SEM}}}(f, f^*) + \text{disp}_{P^{\text{COV}}}(f, f^*) + \mathbb{E}_{P^{\text{COV}}}TV(F_{f^*}||U) \\
&\leq \mathbb{E}_{P_{\text{test}}^{\text{SEM}}}TV(F_f||U) + d_{\mathcal{F}}(P_{\text{test}}^{\text{SEM}}, P^{\text{COV}}) + \lambda.
\end{aligned}
\tag{18}
$$

$\square$

Intuitively speaking, Lemma 1 demonstrates that if the classifier $f$ express high overall uncertainty on $P_{\text{test}}^{\text{SEM}}$ (i.e., the predicted distribution $F_f$ is close to uniform distribution), it will also tend to high uncertain prediction on $P^{\text{COV}}$ given a limited $d_{\mathcal{F}}(P_{\text{test}}^{\text{SEM}}, P^{\text{COV}})$.

**Lemma 2.** *[Inequality between KL and TV] For any K-class distribution $P$ and $Q$ on $\{1, \cdots, K\}$ and $\kappa > 0$, the following inequality holds*

$$\frac{1}{2\kappa}KL(P||Q) + \frac{\kappa}{4} \geq TV(P||Q). \tag{19}$$

*Proof.* The proof can be found in [53] page 8. $\square$

**Lemma 3.** *Denote the OOD detection loss used for MSP detectors as $\mathcal{L}_{\text{reg}}$, then we have*

$$\mathbb{E}_{P_{\text{test}}^{\text{SEM}}}TV(F_f||U) \leq \mathbb{E}_{P_{\text{test}}^{\text{SEM}}}\sqrt{\frac{1}{2}(\mathcal{L}_{\text{reg}}(f) - \log K)} \tag{20}$$

*where $TV(\cdot||\cdot)$ is total variance distance (TVD). $U$ denotes uniform distribution support on $\mathcal{Y} = \{1, 2 \cdots K\}$. $\mathcal{L}_{\text{reg}}$ is defined in [7] is the cross-entropy between predicted distribution $F_f(x)$ and uniform distribution $U$.*

Lemma 2 means that minimizing the OOD detection loss will constrains the predicted distribution to be close to uniform distribution, which is a intuitive and straightforward result.

*Proof.* In $K$-classes classification task, for any sample $\tilde{x}$ drawn from $P_{\mathcal{X}}^{\text{SEM}}$, we have

$$\mathcal{L}_{\text{reg}}(f(\tilde{x})) = KL(U||F_f) + H(U). \tag{21}$$

Applying Pinsker's Inequality, the following inequality holds

$$\mathcal{L}_{\text{reg}}(f(\tilde{x})) = KL(F_f||U) + H(U) \geq 2TV(F_f(\tilde{x})||U)^2 + H(U). \tag{22}$$

Noted that $H(U) = \log K$, we can re-write above inequality as

$$TV(F_f(\tilde{x})||U) \leq \sqrt{\frac{1}{2}(\mathcal{L}_{\text{reg}}(f(\tilde{x})) - \log K)}. \tag{23}$$

Then, by taking expectation on $P_{\mathcal{X}}^{\text{SEM}}$ we can get the result. $\square$

Now we are ready to present the proof of Theorem 1.

*Proof.* By the definition, the generalization error can be written as

$$\text{GError}_{P^{\text{COV}}_{\mathcal{X}\mathcal{Y}}}(f) := \mathbb{E}_{(x,y)\sim P^{\text{COV}}_{\mathcal{X}\mathcal{Y}}} \mathcal{L}_{\text{CE}}(f(x), y) \tag{24}$$
$$= \mathbb{E}_{(x,y)\sim P^{\text{COV}}_{\mathcal{X}\mathcal{Y}}}[KL(P_{true}||F_f(x)) + H(P_{true})]$$

where $P_{true}$ is the target distribution given input $x$ (i.e., the true class distribution) and $\mathcal{L}_{\text{CE}}(\cdot)$ denotes the cross-entropy loss.

Applying Lemma 2, for any $x$ we have

$$KL(P_{true}||F_f) \geq \frac{1}{2\kappa}(TV(P_{true}||F_f)) + \frac{\kappa}{4}. \tag{25}$$

By the sub-additivity of TVD, we have

$$KL(P_{true}||F_f(x)) \geq \frac{1}{2\kappa}(TV(P_{true}||F_f)) + \frac{\kappa}{4} \tag{26}$$
$$\geq \frac{1}{2\kappa}[TV(P_{true}||U) - TV(F_f(x)||U)] + \frac{\kappa}{4}.$$

Taking expectation on $P^{\text{COV}}_{\mathcal{X}\mathcal{Y}}$, we have

$$\text{GError}_{P^{\text{COV}}_{\mathcal{X}\mathcal{Y}}}(f) = \mathbb{E}_{(x,y)\sim P^{\text{COV}}_{\mathcal{X}\mathcal{Y}}} KL(P_{true}||F_f(x)) + H(P_{true})$$
$$\geq \frac{1}{2\kappa}\mathbb{E}_{(x,y)\sim P^{\text{COV}}_{\mathcal{X}\mathcal{Y}}}[TV(P_{true}||U) - TV(F_f(x)||U)] + \frac{\kappa}{4} + \mathbb{E}_{P^{\text{COV}}_{\mathcal{X}\mathcal{Y}}} H(P_{true}) \tag{27}$$

Applying Lemma 1 and Lemma 2,

$$\text{GError}_{P^{\text{COV}}_{\mathcal{X}}}(f) = \mathbb{E}_{P^{\text{COV}}_{\mathcal{X}\mathcal{Y}}} KL[P_{true}||F_f(x)] + \mathbb{E}_{P^{\text{COV}}_{\mathcal{X}\mathcal{Y}}} H(P_{true})$$
$$\geq \frac{1}{2\kappa}\mathbb{E}_{P^{\text{COV}}_{\mathcal{X}\mathcal{Y}}}[TV(P_{true}||U) - TV(F_f(x)||U)] + \frac{\kappa}{4} + \mathbb{E}_{P^{\text{COV}}_{\mathcal{X}\mathcal{Y}}} H(P_{true})$$
$$\geq \frac{1}{2\kappa}\mathbb{E}_{P^{\text{COV}}_{\mathcal{X}\mathcal{Y}}}TV(P_{true}||U) - \frac{1}{2\kappa}\mathbb{E}_{P^{\text{COV}}_{\mathcal{X}\mathcal{Y}}}TV(F_f||U) + \frac{\kappa}{4} + \mathbb{E}_{P^{\text{COV}}_{\mathcal{X}\mathcal{Y}}} H(P_{true})$$
$$\geq \frac{1}{2\kappa}\mathbb{E}_{P^{\text{COV}}_{\mathcal{X}\mathcal{Y}}}TV(P_{true}||U) - \frac{1}{2\kappa}\mathbb{E}_{P^{\text{SEM}}_{\mathcal{X}}}TV(F_f||U)$$
$$\quad - \frac{1}{2\kappa}d_{\mathcal{F}}(P^{\text{COV}}_{\mathcal{X}\mathcal{Y}}, P^{\text{SEM}}_{\mathcal{X}}) - \frac{\lambda}{2\kappa} + \frac{\kappa}{4} + \mathbb{E}_{P^{\text{COV}}_{\mathcal{X}\mathcal{Y}}} H(P_{true})$$
$$\geq \frac{1}{2\kappa}\mathbb{E}_{P^{\text{COV}}_{\mathcal{X}\mathcal{Y}}}TV(P_{true}||U) - \frac{1}{2\kappa}\mathbb{E}_{P^{\text{SEM}}_{\mathcal{X}}}\sqrt{\frac{1}{2}(\mathcal{L}_{\text{reg}}(f) - \log K)}$$
$$\quad - \frac{1}{2\kappa}d_{\mathcal{F}}(P^{\text{COV}}_{\mathcal{X}}, P^{\text{SEM}}_{\mathcal{X}}) - \frac{1}{2\kappa}\lambda + \frac{4}{\kappa} + \mathbb{E}_{P^{\text{COV}}_{\mathcal{X}\mathcal{Y}}} H(P_{true}). \tag{28}$$

Given the fact that $P^{\text{COV}}_{\mathcal{X}\mathcal{Y}}$, $P_{true}$ are both fixed, $H(P_{true})$ and $TV(P_{true}||U)$ are constants for each $x$. Finally, we get

$$\text{GError}_{P^{\text{COV}}_{\mathcal{X}\mathcal{Y}}}(f) \geq C - \frac{1}{2\kappa}\mathbb{E}_{P^{\text{SEM}}_{\mathcal{X}}}\sqrt{\frac{1}{2}(\mathcal{L}_{\text{reg}}(f) - \log K)} - \frac{1}{2\kappa}d_{\mathcal{F}}(P^{\text{COV}}_{\mathcal{X}}, P^{\text{SEM}}_{\mathcal{X}}), \tag{29}$$

where $C = \frac{1}{2\kappa}(\mathbb{E}_{P^{\text{COV}}_{\mathcal{X}}}TV(P_{true}||U) - \lambda + 8) + \mathbb{E}_{P^{\text{COV}}_{\mathcal{X}\mathcal{Y}}} H(P_{true})$ is a constant. $\square$

# B Experimental Details

## B.1 Datasets details

**ID datasets $P^{\text{ID}}$.** ID datasets are chosen following common practice in OOD detection. We use CIFAR-10, CIFAR-100 and ImageNet-200 as $P^{\text{ID}}$. ImageNet-200 is a subset of the original ImageNet-1K introduced by [20, 5].

**Auxiliary OOD datasets $P^{\text{SEM}}_{\text{train}}$.** For CIFAR experiments, we use ImageNet-RC and TIN-597 as auxiliary datasets. ImageNet-RC is a down-sampled variant of the ImageNet-1K, which consists

of 1000 classes and 1,281,167 images. We also conduct experiments on TIN-597 as an alternative for ImageNet-RC. TIN-597 is introduced by recent work [5]. The resolutions of ID and auxiliary samples are both $64 \times 64$. For ImageNet experiments, we use a subset of ImageNet-1K consisting of 200 classes as ID datasets. The remaining images belong to other 800 classes are utilized as auxiliary datasets. The resolutions of ID and auxiliary images are both $224 \times 224$.

**OOD detection test datasets** $P_{\text{test}}^{\text{SEM}}$**.** In CIFAR experiments, following standard practice [8], we use SVHN [40], Textures [42], Places365 [41], iSUN [44], LSUN-C and LSUN-R [43] to evaluate the OOD detection performance. ○ The SVHN test set comprises 26,032 color images of house numbers. ○ Textures (Describable Textures Dataset, DTD) consists of 5,640 images depicting natural textures. ○ Places365 dataset consists scenic images of 365 different categories. Each class consists of 900 images. ○ The iSUN dataset is a subset of the SUN database with 8,925 images. ○ The Large-scale Scene Understanding dataset (LSUN) comprises a testing set with 10,000 images of 10 different scenes. LSUN offers two datasets, LSUN-C and LSUN-R. In LSUN-C, the original high-resolution images are randomly cropped into $32 \times 32$. Meanwhile, in LSUN-R, the images are resized to $32 \times 32$. In ImageNet experiments, we follow the settings of [5], where OpenImage-O [54], SSB-hard [55], Textures [42], iNaturalist [45] and NINCO [47] are selected as OOD detection test datasets. ○ OpenImage-O contains 17632 manually filtered images and is $7.8 \times$ larger than the ImageNet dataset. ○ SSB-hard is selected from ImageNet-21K. It consists of 49K images and 980 categories. ○ Textures (Describable Textures Dataset, DTD) consists of 5,640 images depicting natural textures. ○ iNaturalist consists of 859000 images from over 5000 different species of plants and animals. ○ NINCO consists with a total of 5879 samples of 64 classes which are non-overlapped with ImageNet-1K.

**OOD generalization test datasets** $P^{\text{COV}}$**.** Following previous work [4], we corrupt the original test data with Gaussian noise of zero mean and variance of 5 in the main paper. In appendix, we conduct additional experiments involving CIFAR10-C, CIFAR100-C and ImageNet-C [56] with 15 diverse types of noise.

## B.2   Implementation details

### B.2.1   CIFAR experiments.

We use WideResNet-40-10 [57] as the backbone network, which comprises 40 layers. The widen factor is set to 10. We use SGD optimizer to train all methods with dropout strategy. The dropout rate is 0.3. The momentum is set to 0.9 and weight decay is set to 0.0005.

**Pretraining details.** The pretrained model is obtained by training WideResNet-40-10 for 200 epochs with an initial learning rate of 0.1. We decay the learning rate by a factor of 0.2 at the 60-th, 120-th, and 160-th epochs. Batch size is set to 128.

**Finetuning details.** ○ For Entropy and EBM, we finetune the pretrained model for 20 epochs with an initial learning rate of 0.001, utilizing a cosine annealing strategy to adjust the learning rate. Following the official implementation, the weight of OOD detection regularization term is set to 0.5 and 0.1 for Entropy and EBM (finetune) respectively. The hyperparameters $m_{\text{ID}}$ and $m_{\text{OOD}}$ in EBM regularization learning are set to -25 and -7 respectively. The ID batch size is 128 and the OOD batch size is set to 256. ○ For SCONE, we finetune the pretrained model for 10 epochs with an initial learning rate of 0.0002, utilizing a cosine annealing strategy to adjust the learning rate. The batch size is 32, the OOD batch size is 64. The margin of the OOD detection boundary is set to 1. To be aligned with most previous works in OOD detection and generalization, we assume $P_{\mathcal{X}}^{\text{COV}}$ is unavailable during finetuning. ○ For WOODS, we finetune the pretrained model for 10 epochs with an initial learning rate of 0.0002, utilizing a cosine annealing strategy to adjust the learning rate. The ID batch size is 32, the OOD batch size is 64. Other hyperparameters settings are consistent with SCONE. ○ For DUL, $\alpha_0$ is set to 12. While finetuning on CIFAR10, the $m_{\text{ID}}$ and $m_{\text{OOD}}$ are set to 10 and 30 respectively. The weight $\lambda, \gamma$ are set to 0.3 and 2. We train for 20 epochs with an initial learning rate of 0.00005, utilizing a cosine annealing strategy to adjust the learning rate. While finetuning on CIFAR100/TIN-597, the $m_{\text{ID}}$ and $m_{\text{OOD}}$ are set to 10 and 30 respectively. The weights $\lambda, \gamma$ are set to 0.05 and 2 respectively. We finetune for 20 epochs with an initial learning rate of 0.00005, utilizing a cosine annealing strategy to adjust the learning rate. While finetuning on CIFAR100/ImageNet-RC, we set $h_0 = 0$. The $m_{\text{ID}}$ and $m_{\text{OOD}}$ are set to -430 and -370 respectively. We train for 30 epochs with an initial learning rate of 0.0001, utilizing a cosine annealing strategy to adjust the learning rate.

The weights $\lambda, \gamma$ are set to 0.1 and 1 respectively. For CIFAR-100/ImageNet-RC, we set $\tau = 2$ and otherwise $\tau = 1$. ∘ For DUL$^\dagger$, we use Thompson sampling strategy [11] for OOD informativeness mining. The sampling hyperparameters are consistent with that of POEM.

**Training from scratch details.** ∘ For POEM, we train from scratch for 200 epochs with an initial learning rate of 0.1, and decay the learning rate by a factor of 0.2 at the 60-th, 120-th, and 160-th epochs following [57]. The ID and OOD batch size are set to 128 and 256 respectively. Following the official implementation, the pool of outliers consists of randomly selected 400,000 samples from auxiliary datasets, and only 50,000 samples (same size as the ID training set) are selected for training based on the boundary score. ∘ For DPN, we train for 200 epochs with an initial learning rate of 0.1, and decay the learning rate by a factor of 0.2 at the 60-th, 120-th, and 160-th epochs. The Dirichlet parameters $\alpha$ are calculated by performing ReLU plus one on the model's outputs, i.e., $\alpha = \mathrm{ReLU}(f(x)) + 1$. $\alpha_0$ is set to 15 and 12 respectively when training on CIFAR10 and CIFAR100. The auxiliary datasets are ImageNet and TIN-597. The ID and OOD batch size are set to 128 and 256 respectively. When training on CIFAR100/TIN-597, the OOD regularization weight $\lambda$ is set to 0.05. In other cases, $\lambda$ is set to 0.5.

### B.2.2 ImageNet experiments.

We use ResNet18 [58] as the backbone network. We use SGD optimizer to train all the models. The momentum is set to 0.9.

**Pretraining details.** The pretrained model is obtained by training ResNet18 for 100 epochs with an initial learning rate of 0.1, utilizing a cosine annealing strategy to adjust the learning rate. The weight decay is set to 0.0001. Batch size is set to 64.

**Finetuning details.** ∘ For Energy regularized learning, we finetune the pretrained model for 10 epochs with an initial learning rate of 0.001, utilizing a cosine annealing strategy to adjust the learning rate. The weight decay is set to 0.0001. Following the official implementation, the weights of OOD detection regularization term are set to 0.1. Specifically, the $m_{\mathrm{ID}}$ and $m_{\mathrm{OOD}}$ in energy regularization method are set to -25 and -7 respectively. The ID batch size is 64 and the OOD batch size is set to 128. ∘ For Entropy, we finetune the pretrained model for 10 epochs with an initial learning rate of 0.001, utilizing a cosine annealing strategy to adjust the learning rate. The weight decay is set to 0.0001. The ID and OOD batch size are set to 64 and 128 respectively. Following the official implementation, the weights of OOD detection regularization term are set to 0.5.∘ For SCONE, we finetune the pretrained model for 10 epochs with an initial learning rate of 0.0002, utilizing a cosine annealing strategy to adjust the learning rate. The weight decay is set to 0.0005. The batch size is 32, the OOD batch size is 64. The margin of the OOD detection boundary is set to 1. To be aligned with most previous works in OOD detection and generalization, we assume $P_{\mathcal{X}}^{\mathrm{COV}}$ is unavailable during finetuning. ∘ For WOODS, we finetune the pretrained model for 10 epochs with an initial learning rate of 0.0002, utilizing a cosine annealing strategy to adjust the learning rate. The batch size is 32, the OOD batch size is 64. Other hyperparameters of WOODS are consistent with SCONE. ∘ For our DUL, the Dirichlet parameters $\alpha$ are calculated by performing ReLU and exp operation on the model's outputs, i.e., $\alpha = \exp(\mathrm{ReLU}(f(x)))$. For numerical stability on large scale benchmark, we measure the distributional uncertainty by the strength of Dirichlet distribution. $\lambda, \gamma$ are set to 0.1 and 4 respectively. We set $\tau = 1$ in large-scale ImageNet experiments.

**Training from scratch details.** ∘ For DPN, we train ResNet18 for 100 epochs with an initial learning rate of 0.1, utilizing a cosine annealing strategy to adjust the learning rate. The weight decay is set to 0.0001. Batch size is set to 64. The Dirichlet parameters $\alpha$ are calculated by performing ReLU plus one on the model's outputs, i.e., $\alpha = \mathrm{ReLU}(f(x)) + 1$. The ID classification loss is set to KL-divergence between predicted class distribution under Dirichlet prior and target distribution because of the inconvenience of directly setting $\alpha_0$. The target distribution is obtained by label smoothing strategy with parameter of 0.01 [9]. The weight of regularization term applied on OOD auxiliary samples is 1.

**Algorithm 1:** Pseudo Code of Decoupled Uncertainty Learning (DUL)

**Input** : ID data $P^{\text{ID}}$, auxiliary outliers $P^{\text{SEM}}_{\text{train}}$, classifier $f_{\theta_0}$ pretrained on $P^{\text{ID}}$.
**Output :** finetuned classifier $f_\theta$

1 Initialize $\theta = \theta_0$;
2 **for** *each iteration* **do**
3     Obtain ID sample $(x, y)$ from $P^{\text{ID}}$ and auxiliary outlier $\tilde{x}$ from $P^{\text{SEM}}_{\text{train}}$;
4     Update model parameters $\theta$ by minimizing objective defined in Eq. 12;
5 **end**

## C Additional Results

### C.1 Uncertainty estimation.

We add Gaussian noise with zero mean and varying variance $\epsilon$ on CIFAR-10 and investigate the estimated distributional uncertainty and overall uncertainty. Distributional uncertainty is measured by differential entropy. It clear that with DUL regularization, the prediction yields a low overall uncertainty and high distributional uncertainty on covariate-shifted data. We conduct experiments on CIFAR-10/ImageNet-RC and CIFAR-10/TIN-597, tabular results are shown in Tab. 4.

Table 4: Mean value of estimated uncertainty on CIFAR-10-C with varying severity of Gaussian noise with zero mean and variance of $\epsilon$.

| $\mathcal{P}^{\text{in}}_{\mathcal{X}}/\mathcal{P}^{\text{aux}}_{\mathcal{X}}$ | Uncertainty type | DUL | $\epsilon = 0.0$ | $\epsilon = 2.0$ | $\epsilon = 4.0$ | $\epsilon = 6.0$ | $\epsilon = 8.0$ | $\epsilon = 10.0$ |
|---|---|---|---|---|---|---|---|---|
| CIFAR-10 ImageNet-RC | Distributional uncertainty | ✗ | -21.33 | -18.94 | -15.62 | -13.71 | -12.91 | -12.85 |
| | | ✓ | -21.23 | -19.42 | -16.96 | -15.47 | -14.85 | -14.58 |
| | Total uncertainty | ✗ | 0.04 | 0.47 | 1.31 | 1.93 | 2.20 | 2.28 |
| | | ✓ | 0.03 | 0.17 | 0.48 | 0.77 | 0.98 | 1.14 |
| CIFAR-10 TIN-597 | Distributional uncertainty | ✗ | -20.94 | -20.14 | -18.29 | -16.23 | -14.71 | -13.72 |
| | | ✓ | -21.48 | -20.68 | -19.21 | -17.70 | -16.57 | -15.78 |
| | Total uncertainty | ✗ | 0.06 | 0.15 | 0.51 | 1.06 | 1.55 | 1.92 |
| | | ✓ | 0.04 | 0.08 | 0.18 | 0.34 | 0.51 | 0.68 |

### C.2 Time-consuming comparison.

We compare the time-cost of proposed DUL to other training-required OOD detection methods in Tab. 5. We run all the experiments on one single NVIDIA GeForce RTX-3090 GPU. Compared with other OOD detection methods in a finetune manner, DUL does not introduce noticeably extra cost of computation.

Table 5: Average execution times (s) per epoch of training required OOD detection methods. Compare to other OOD detection methods, DUL does not introduce noticeable computational cost.

| Method | CIFAR-10/ImageNet-RC | CIFAR-10/TIN-597 | CIFAR-100/ImageNet-RC | CIFAR-100/TIN-597 |
|---|---|---|---|---|
| EBM (finetune) | 354.87 | 229.12 | 355.70 | 112.47 |
| Entropy | 355.98 | 140.88 | 1108.43 | 120.85 |
| DPN | 842.05 | 75.62 | 841.69 | 80.98 |
| POEM | 615.51 | 483.04 | 825.87 | 500.01 |
| WOODS | 808.77 | 291.37 | 906.67 | 282.50 |
| SCONE | 911.83 | 183.57 | 1169.33 | 160.00 |
| DUL | 329.58 | 101.08 | 925.62 | 100.43 |
| DUL* | 598.68 | 597.82 | 544.26 | 595.25 |

### C.3 Full results with standard deviation.

Full results with standard deviation are presented this section. In CIFAR experiments, we report the mean and standard deviation in 5 random runs. In ImageNet experiments, we report the mean and

standard deviation in 3 random runs to be consist with [5]. CIFAR experimental results are shown in Tab. 6. Large-scale ImageNet results are shown in Tab. 7.

Table 6: OOD detection and generalization performance comparison with standard variance. Marginal improvement and degradation ($\geq 0.5$) compare to the baseline method MSP are highlighted in blue or red respectively. The **best** and second best results are in bold or underlined. DUL is the only method achieves state-of-art OOD detection performance (mostly the best or second best) without trade-offs on generalization i.e., the value of entire row is either blue or black.

| $\mathcal{P}^{\mathrm{ID}}/\mathcal{P}^{\mathrm{SEM}}_{\mathrm{train}}$ | Method | ID/OOD generalization | | OOD detection | | |
| --- | --- | --- | --- | --- | --- | --- |
| | | ID-Acc. ↑ | OOD-Acc. ↑ | FPR ↓ | AUROC ↑ | AUPR ↑ |
| CIFAR-10 Only | MSP | $96.11^{\pm 0.09}$ | $87.35^{\pm 0.58}$ | $41.96^{\pm 3.85}$ | $89.28^{\pm 1.12}$ | $68.00^{\pm 2.19}$ |
| | EBM (pretrain) | $96.11^{\pm 0.09}$ | $87.35^{\pm 0.58}$ | $32.45^{\pm 3.45}$ | $89.34^{\pm 1.21}$ | $75.22^{\pm 2.67}$ |
| | Maxlogits | $96.11^{\pm 0.09}$ | $87.35^{\pm 0.58}$ | $32.90^{\pm 3.51}$ | $89.26^{\pm 1.21}$ | $74.47^{\pm 2.55}$ |
| | Mahalanobis | $96.11^{\pm 0.09}$ | $87.35^{\pm 0.58}$ | $32.53^{\pm 9.61}$ | $93.93^{\pm 2.68}$ | $74.96^{\pm 7.47}$ |
| CIFAR-10 ImageNet-RC | Entropy | $96.04^{\pm 0.14}$ | $72.57^{\pm 3.87}$ | $6.63^{\pm 0.80}$ | $98.72^{\pm 0.14}$ | $94.00^{\pm 1.00}$ |
| | EBM (pretrain) | $\mathbf{96.10}^{\pm 0.23}$ | $79.03^{\pm 2.53}$ | $\underline{3.61}^{\pm 0.71}$ | $98.39^{\pm 0.39}$ | $94.88^{\pm 0.91}$ |
| | POEM | $94.32^{\pm 0.14}$ | $78.89^{\pm 2.25}$ | $\mathbf{3.32}^{\pm 0.41}$ | $\mathbf{98.99}^{\pm 0.17}$ | $\mathbf{99.38}^{\pm 0.12}$ |
| | DPN | $95.69^{\pm 0.17}$ | $85.52^{\pm 0.51}$ | $4.28^{\pm 0.60}$ | $98.53^{\pm 0.17}$ | $94.93^{\pm 0.60}$ |
| | WOODS | $96.01^{\pm 0.16}$ | $80.14^{\pm 1.69}$ | $7.12^{\pm 1.54}$ | $98.40^{\pm 0.21}$ | $92.92^{\pm 0.96}$ |
| | SCONE | $95.96^{\pm 0.08}$ | $78.80^{\pm 1.57}$ | $7.02^{\pm 1.06}$ | $98.45^{\pm 0.12}$ | $92.46^{\pm 0.93}$ |
| | DUL (ours) | $96.02^{\pm 0.07}$ | $\mathbf{88.01}^{\pm 0.54}$ | $5.89^{\pm 0.35}$ | $98.47^{\pm 0.12}$ | $92.44^{\pm 1.14}$ |
| | DUL$^\dagger$ (ours) | $\underline{96.04}^{\pm 0.03}$ | $\underline{87.53}^{\pm 0.70}$ | $5.99^{\pm 0.25}$ | $98.28^{\pm 0.11}$ | $\underline{98.40}^{\pm 0.36}$ |
| CIFAR-10 TIN-597 | Entropy | $\underline{95.94}^{\pm 0.00}$ | $80.51^{\pm 0.68}$ | $11.60^{\pm 0.82}$ | $97.93^{\pm 0.15}$ | $92.16^{\pm 0.50}$ |
| | EBM (pretrain) | $95.38^{\pm 0.13}$ | $83.67^{\pm 1.41}$ | $19.36^{\pm 1.92}$ | $87.51^{\pm 1.53}$ | $83.63^{\pm 1.73}$ |
| | POEM | $95.44^{\pm 0.18}$ | $83.17^{\pm 1.39}$ | $24.34^{\pm 2.48}$ | $86.83^{\pm 1.13}$ | $94.25^{\pm 0.53}$ |
| | DPN | $94.39^{\pm 0.38}$ | $79.23^{\pm 2.95}$ | $17.27^{\pm 1.07}$ | $94.92^{\pm 0.65}$ | $87.67^{\pm 0.88}$ |
| | WOODS | $95.57^{\pm 0.64}$ | $83.12^{\pm 1.71}$ | $\underline{7.58}^{\pm 0.52}$ | $\mathbf{98.29}^{\pm 0.04}$ | $93.39^{\pm 0.39}$ |
| | SCONE | $95.19^{\pm 0.77}$ | $84.68^{\pm 1.44}$ | $8.02^{\pm 0.92}$ | $98.22^{\pm 0.08}$ | $93.08^{\pm 0.30}$ |
| | DUL (ours) | $\mathbf{96.06}^{\pm 0.08}$ | $\underline{87.93}^{\pm 0.62}$ | $\mathbf{6.87}^{\pm 0.82}$ | $98.21^{\pm 0.12}$ | $91.29^{\pm 1.18}$ |
| | DUL$^\dagger$ (ours) | $\underline{95.94}^{\pm 0.09}$ | $\mathbf{88.10}^{\pm 0.27}$ | $10.34^{\pm 0.34}$ | $97.67^{\pm 0.09}$ | $\mathbf{98.59}^{\pm 0.24}$ |
| CIFAR-100 Only | MSP | $80.99^{\pm 0.16}$ | $55.95^{\pm 1.38}$ | $74.63^{\pm 2.43}$ | $80.19^{\pm 1.65}$ | $42.59^{\pm 2.79}$ |
| | EBM (pretrain) | $80.99^{\pm 0.16}$ | $55.95^{\pm 1.38}$ | $67.42^{\pm 4.35}$ | $82.67^{\pm 1.82}$ | $49.35^{\pm 4.00}$ |
| | Maxlogits | $80.99^{\pm 0.16}$ | $55.95^{\pm 1.38}$ | $69.32^{\pm 3.97}$ | $82.30^{\pm 1.79}$ | $47.60^{\pm 3.68}$ |
| | Mahalanobis | $80.99^{\pm 0.16}$ | $55.95^{\pm 1.38}$ | $61.51^{\pm 3.62}$ | $85.97^{\pm 1.22}$ | $56.10^{\pm 3.22}$ |
| CIFAR-100 ImageNet-RC | Entropy | $80.21^{\pm 0.09}$ | $45.48^{\pm 0.78}$ | $22.29^{\pm 1.32}$ | $95.33^{\pm 0.28}$ | $82.34^{\pm 1.11}$ |
| | EBM (finetune) | $80.53^{\pm 0.22}$ | $48.14^{\pm 0.33}$ | $13.47^{\pm 0.43}$ | $96.78^{\pm 0.13}$ | $87.84^{\pm 0.86}$ |
| | POEM | $78.15^{\pm 0.18}$ | $42.18^{\pm 2.34}$ | $\mathbf{9.89}^{\pm 0.36}$ | $\mathbf{97.79}^{\pm 0.12}$ | $\mathbf{98.40}^{\pm 0.08}$ |
| | DPN | $78.90^{\pm 0.25}$ | $50.14^{\pm 0.36}$ | $18.36^{\pm 0.82}$ | $95.42^{\pm 0.17}$ | $74.45^{\pm 18.40}$ |
| | WOODS | $80.69^{\pm 0.30}$ | $54.38^{\pm 4.42}$ | $38.15^{\pm 12.91}$ | $92.01^{\pm 3.23}$ | $71.79^{\pm 7.98}$ |
| | SCONE | $80.80^{\pm 0.30}$ | $\mathbf{56.73}^{\pm 4.66}$ | $47.60^{\pm 14.73}$ | $89.61^{\pm 3.75}$ | $65.29^{\pm 9.66}$ |
| | DUL (ours) | $\mathbf{81.30}^{\pm 0.19}$ | $\underline{56.27}^{\pm 1.82}$ | $12.49^{\pm 0.22}$ | $95.24^{\pm 0.09}$ | $86.72^{\pm 0.76}$ |
| | DUL$^\dagger$ (ours) | $\underline{81.22}^{\pm 0.23}$ | $56.07^{\pm 0.54}$ | $\underline{11.75}^{\pm 1.69}$ | $95.33^{\pm 0.79}$ | $\underline{96.45}^{\pm 0.42}$ |
| CIFAR-100 TIN-597 | Entropy | $80.15^{\pm 0.17}$ | $46.25^{\pm 1.42}$ | $26.88^{\pm 2.06}$ | $93.50^{\pm 0.36}$ | $79.81^{\pm 1.31}$ |
| | EBM (finetune) | $79.94^{\pm 0.27}$ | $50.00^{\pm 0.93}$ | $26.87^{\pm 1.15}$ | $91.68^{\pm 0.45}$ | $80.08^{\pm 0.76}$ |
| | POEM | $78.68^{\pm 0.13}$ | $52.53^{\pm 1.06}$ | $32.71^{\pm 0.96}$ | $91.30^{\pm 0.68}$ | $94.65^{\pm 0.49}$ |
| | DPN | $78.44^{\pm 0.22}$ | $47.67^{\pm 0.28}$ | $\underline{25.02}^{\pm 2.19}$ | $\mathbf{93.55}^{\pm 0.19}$ | $81.63^{\pm 1.27}$ |
| | WOODS | $79.26^{\pm 2.28}$ | $53.13^{\pm 2.97}$ | $36.71^{\pm 7.92}$ | $92.15^{\pm 2.35}$ | $73.42^{\pm 4.72}$ |
| | SCONE | $79.53^{\pm 1.82}$ | $52.70^{\pm 0.96}$ | $35.60^{\pm 9.50}$ | $92.47^{\pm 2.19}$ | $73.58^{\pm 5.21}$ |
| | DUL (ours) | $\mathbf{80.85}^{\pm 0.24}$ | $\underline{56.19}^{\pm 1.53}$ | $23.32^{\pm 1.11}$ | $\mathbf{94.48}^{\pm 0.35}$ | $80.82^{\pm 1.62}$ |
| | DUL$^\dagger$ (ours) | $\underline{80.50}^{\pm 0.25}$ | $\mathbf{56.22}^{\pm 1.29}$ | $\mathbf{22.75}^{\pm 0.88}$ | $90.88^{\pm 0.28}$ | $\mathbf{96.33}^{\pm 0.09}$ |

## C.4 Results of different types of corruption.

We conduct additional experiments on CIFAR10-C, CIFAR100-C and ImageNet-C with 15 different types of corruption. The results validate that the proposed method can improve the overall performance under different types of corruption.

## C.5 OOD detection results on individual datasets.

We provide OOD detection results of DUL on each individual OOD detection test dataset in Tab. 8 and Tab. 9, based on the checkpoint with random seed 1.

Table 7: OOD detection and generalization performance comparison with standard variance. Substantially improvement and degradation ($\geq 0.5$) compare to baseline method w.r.t. MSP are highlighted in blue or red respectively. The **best** and second best results are in bold or underlined. Similar with CIFAR experiments, DUL establishes strong OOD detection performance (always the best or second best) without degraded generalization i.e., the entire row is either blue or black.

| $\mathcal{P}^{ID}/\mathcal{P}^{SEM}_{train}$ | Method | ID/OOD Generalization | | OOD Detection | | |
|---|---|---|---|---|---|---|
| | | ID-Acc. ↑ | OOD-Acc. ↑ | FPR ↓ | AUROC ↑ | AUPR ↑ |
| ImageNet-200

ImageNet-800 | MSP | $85.15^{\pm0.33}$ | $74.84^{\pm0.47}$ | $58.23^{\pm1.54}$ | $86.98^{\pm0.24}$ | $82.27^{\pm0.32}$ |
| | EBM (pretrain) | $85.15^{\pm0.33}$ | $74.84^{\pm0.47}$ | $51.94^{\pm0.82}$ | $88.18^{\pm0.11}$ | $84.75^{\pm0.08}$ |
| | Maxlogits | $85.15^{\pm0.33}$ | $74.84^{\pm0.47}$ | $51.62^{\pm0.20}$ | $88.30^{\pm0.09}$ | $84.71^{\pm0.07}$ |
| | Entropy | $84.92^{\pm0.30}$ | $74.75^{\pm0.52}$ | $53.62^{\pm0.76}$ | $89.05^{\pm0.01}$ | $85.02^{\pm0.08}$ |
| | EBM (finetune) | $84.14^{\pm0.11}$ | $73.31^{\pm0.57}$ | $59.73^{\pm0.83}$ | $87.54^{\pm0.03}$ | $82.81^{\pm0.16}$ |
| | DPN | $84.87^{\pm0.30}$ | $74.40^{\pm0.90}$ | $63.84^{\pm0.70}$ | $87.18^{\pm0.18}$ | $80.69^{\pm0.35}$ |
| | WOODS | $84.99^{\pm0.62}$ | $74.98^{\pm0.46}$ | $51.71^{\pm2.84}$ | $88.30^{\pm0.56}$ | $84.80^{\pm0.98}$ |
| | SCONE | $84.93^{\pm0.71}$ | $74.91^{\pm0.49}$ | $52.52^{\pm3.54}$ | $88.19^{\pm0.41}$ | $84.50^{\pm1.08}$ |
| | DUL (ours) | **$85.65^{\pm0.07}$** | **$75.59^{\pm0.12}$** | **$49.14^{\pm0.13}$** | **$89.27^{\pm0.03}$** | **$85.62^{\pm0.03}$** |

Table 8: OOD detection results of DUL on each individual OOD detection test dataset.

| $\mathcal{P}^{ID}/\mathcal{P}^{SEM}_{train}$ | Method | LSUN-crop | | Places365 | | LSUN-resize | | iSUN | | Texture | | SVHN | |
|---|---|---|---|---|---|---|---|---|---|---|---|---|---|
| | | FPR↓ | AUROC↑ | FPR↓ | AUROC↑ | FPR↓ | AUROC↑ | FPR↓ | AUROC↑ | FPR↓ | AUROC↑ | FPR↓ | AUROC↑ |
| CIFAR-10
ImageNet-RC | DUL
DUL$^\dagger$ | 6.75
12.79 | 98.75
98.08 | 15.55
14.99 | 96.34
96.11 | 0.00
0.00 | 99.63
99.49 | 0.00
0.00 | 99.57
99.45 | 3.20
1.72 | 98.89
98.79 | 8.35
4.84 | 98.24
98.47 |
| CIFAR-10
TIN-597 | DUL
DUL$^\dagger$ | 0.90
3.48 | 99.47
99.25 | 22.40
32.81 | 95.21
91.48 | 0.10
0.00 | 99.52
99.78 | 0.30
0.00 | 99.45
99.78 | 8.00
13.97 | 97.94
97.15 | 5.90
9.69 | 98.52
98.28 |
| CIFAR-100
ImageNet-RC | DUL
DUL$^\dagger$ | 44.65
47.75 | 83.73
81.29 | 21.05
14.92 | 79.30
95.08 | 0.00
0.00 | 99.63
99.58 | 0.00
0.00 | 99.61
99.46 | 3.05
1.70 | 98.53
98.70 | 5.00
4.44 | 98.15
98.02 |
| CIFAR-100
TIN-597 | DUL
DUL$^\dagger$ | 7.05
6.25 | 98.69
98.55 | 65.35
78.95 | 83.60
64.09 | 3.80
0.02 | 99.03
99.84 | 3.50
0.00 | 98.98
99.83 | 34.45
39.79 | 92.21
85.02 | 21.15
5.61 | 95.81
98.51 |

Table 9: OOD detection results of DUL on each OOD detection test dataset.

| $\mathcal{P}^{ID}/\mathcal{P}^{SEM}_{train}$ | Method | OpenImage-O | | SSB-hard | | Textures | | iNaturalist | | NINCO | |
|---|---|---|---|---|---|---|---|---|---|---|---|
| | | FPR↓ | AUROC↑ | FPR↓ | AUROC↑ | FPR↓ | AUROC↑ | FPR↓ | AUROC↑ | FPR↓ | AUROC↑ |
| ImageNet-200/800 | DUL | 49.68 | 91.31 | 72.40 | 80.60 | 30.76 | 92.98 | 33.21 | 94.76 | 59.92 | 86.61 |

Table 10: Classification error rate comparison on CIFAR10-C. ID dataset is CIFAR10.

| Method | $\mathcal{P}^{SEM}_{train}$ | Noise | | | Blur | | | | Weather | | | | Digital | | | Avg. |
|---|---|---|---|---|---|---|---|---|---|---|---|---|---|---|---|---|
| | | Gauss. | Shot | Impul. | Defoc. | Glass | Motion | Zoom | Snow | Frost | Fog | Brit. | Contr. | Elastic | Pixel | JPEG | |
| MSP | None | 53.35 | 39.98 | 44.07 | 15.61 | 42.35 | 19.33 | 19.83 | 14.39 | 18.02 | 9.61 | 5.09 | 18.19 | 13.45 | 22.71 | 18.61 | 23.64 |
| EBM (pretrain) | | 53.35 | 39.98 | 44.07 | 15.61 | 42.35 | 19.33 | 19.83 | 14.39 | 18.02 | 9.61 | 5.09 | 18.19 | 13.45 | 22.71 | 18.61 | 23.64 |
| Maxlogits | | 53.35 | 39.98 | 44.07 | 15.61 | 42.35 | 19.33 | 19.83 | 14.39 | 18.02 | 9.61 | 5.09 | 18.19 | 13.45 | 22.71 | 18.61 | 23.64 |
| Mahalanobis | | 53.35 | 39.98 | 44.07 | 15.61 | 42.35 | 19.33 | 19.83 | 14.39 | 18.02 | 9.61 | 5.09 | 18.19 | 13.45 | 22.71 | 18.61 | 23.64 |
| Entropy | ImageNet-RC | 67.20 | 53.99 | 70.93 | 17.09 | 81.99 | 20.00 | 22.17 | 22.41 | 31.78 | 11.43 | 5.82 | 16.99 | 14.28 | 28.92 | 21.37 | 32.42 |
| EBM (finetune) | | 62.49 | 50.00 | 73.01 | 21.01 | 73.93 | 20.06 | 25.80 | 21.02 | 27.00 | 11.55 | 5.60 | 15.08 | 15.15 | 31.46 | 22.69 | 31.72 |
| DPN | | 50.19 | 38.80 | 59.81 | 16.98 | 52.81 | 19.68 | 22.24 | 18.25 | 20.11 | 11.10 | 5.74 | 18.01 | 14.41 | 24.30 | 18.76 | 26.08 |
| POEM | | 47.52 | 38.16 | 63.39 | 22.64 | 67.37 | 24.12 | 28.43 | 23.44 | 27.33 | 14.30 | 7.65 | 19.02 | 18.37 | 31.87 | 22.23 | 30.40 |
| WOODS | | 62.36 | 49.09 | 60.77 | 15.56 | 74.37 | 19.05 | 20.16 | 18.79 | 25.41 | 9.84 | 5.45 | 16.76 | 14.16 | 24.07 | 20.45 | 29.09 |
| SCONE | | 63.29 | 50.01 | 61.96 | 15.61 | 77.16 | 18.65 | 20.24 | 19.77 | 26.43 | 9.82 | 5.54 | 16.88 | 14.09 | 24.46 | 20.64 | 29.64 |
| DUL (Ours) | | 53.46 | 40.11 | 43.71 | 15.62 | 42.77 | 19.49 | 19.89 | 14.27 | 18.07 | 9.82 | 5.06 | 18.49 | 13.56 | 22.65 | 18.69 | 23.71 |
| Entropy | TIN-597 | 65.68 | 52.08 | 56.49 | 18.68 | 51.96 | 22.73 | 24.43 | 18.57 | 24.64 | 10.56 | 5.51 | 16.38 | 15.88 | 25.19 | 41.25 | 30.00 |
| EBM (finetune) | | 58.95 | 44.89 | 52.91 | 18.30 | 47.25 | 23.99 | 24.33 | 17.70 | 22.18 | 12.23 | 6.07 | 19.78 | 17.11 | 26.85 | 51.31 | 29.59 |
| DPN | | 48.07 | 38.05 | 42.56 | 22.54 | 47.39 | 27.29 | 30.41 | 20.46 | 26.17 | 14.53 | 7.22 | 24.54 | 18.33 | 27.40 | 26.23 | 28.08 |
| POEM | | 51.06 | 39.56 | 47.20 | 16.08 | 48.97 | 20.23 | 21.40 | 17.36 | 22.15 | 11.29 | 5.94 | 18.08 | 14.87 | 22.08 | 31.74 | 25.87 |
| WOODS | | 60.01 | 46.74 | 51.26 | 18.04 | 49.83 | 20.95 | 23.17 | 17.34 | 22.66 | 10.02 | 5.69 | 16.02 | 16.32 | 23.55 | 25.92 | 27.17 |
| SCONE | | 58.20 | 44.92 | 50.36 | 17.14 | 48.47 | 20.64 | 22.80 | 17.05 | 21.15 | 9.87 | 6.07 | 17.24 | 16.17 | 24.89 | 25.53 | 26.70 |
| DUL (Ours) | | 54.56 | 40.84 | 44.18 | 15.67 | 43.61 | 19.93 | 19.74 | 14.49 | 18.34 | 9.84 | 5.18 | 19.24 | 13.95 | 22.91 | 18.66 | 24.08 |

Table 11: Classification error rate comparison on CIFAR100-C. ID dataset is CIFAR100.

| Method | $\mathcal{P}^{\text{SEM}}_{\text{train}}$ | Gauss. | Shot | Impul. | Defoc. | Glass | Motion | Zoom | Snow | Frost | Fog | Brit. | Contr. | Elastic | Pixel | JPEG | Avg. |
|---|---|---|---|---|---|---|---|---|---|---|---|---|---|---|---|---|---|
| MSP | | 76.98 | 67.70 | 74.51 | 36.08 | 78.49 | 40.69 | 41.39 | 40.52 | 46.66 | 31.44 | 22.59 | 40.29 | 35.31 | 42.83 | 44.93 | 48.03 |
| EBM (pretrain) | None | 76.98 | 67.70 | 74.51 | 36.08 | 78.49 | 40.69 | 41.39 | 40.52 | 46.66 | 31.44 | 22.59 | 40.29 | 35.31 | 42.83 | 44.93 | 48.03 |
| Maxlogits | | 76.98 | 67.70 | 74.51 | 36.08 | 78.49 | 40.69 | 41.39 | 40.52 | 46.66 | 31.44 | 22.59 | 40.29 | 35.31 | 42.83 | 44.93 | 48.03 |
| Mahalanobis | | 76.98 | 67.70 | 74.51 | 36.08 | 78.49 | 40.69 | 41.39 | 40.52 | 46.66 | 31.44 | 22.59 | 40.29 | 35.31 | 42.83 | 44.93 | 48.03 |
| Entropy | | 81.69 | 72.89 | 86.99 | 38.19 | 88.94 | 42.20 | 44.62 | 45.50 | 53.90 | 33.14 | 24.40 | 39.77 | 37.19 | 47.62 | 48.82 | 52.39 |
| EBM (finetune) | | 82.33 | 73.51 | 90.58 | 39.20 | 90.71 | 41.43 | 44.40 | 47.29 | 54.88 | 34.26 | 24.72 | 39.15 | 37.65 | 51.47 | 49.29 | 53.39 |
| DPN | | 77.60 | 68.59 | 83.59 | 39.74 | 86.77 | 43.33 | 44.91 | 47.01 | 53.69 | 36.37 | 26.11 | 43.45 | 37.56 | 47.48 | 44.47 | 52.04 |
| POEM | ImageNet-RC | 83.65 | 76.63 | 88.02 | 44.09 | 90.05 | 44.29 | 46.44 | 52.80 | 60.84 | 39.49 | 28.90 | 43.35 | 42.49 | 56.06 | 54.50 | 56.77 |
| WOODS | | 77.51 | 68.48 | 77.47 | 36.38 | 80.62 | 40.92 | 41.93 | 41.27 | 47.08 | 31.36 | 23.30 | 39.74 | 36.21 | 42.87 | 46.16 | 48.75 |
| SCONE | | 76.43 | 67.30 | 75.32 | 36.23 | 77.00 | 40.92 | 41.59 | 40.22 | 45.53 | 31.35 | 23.09 | 40.10 | 35.87 | 42.42 | 45.42 | 47.92 |
| DUL (Ours) | | 77.23 | 67.95 | 75.13 | 35.83 | 78.52 | 39.74 | 40.64 | 39.76 | 46.13 | 30.96 | 22.43 | 39.17 | 34.81 | 43.03 | 44.64 | 47.73 |
| Entropy | | 83.34 | 75.44 | 79.77 | 38.02 | 82.74 | 42.22 | 44.17 | 44.52 | 53.36 | 32.80 | 23.67 | 38.00 | 37.85 | 44.55 | 62.91 | 52.22 |
| EBM (finetune) | | 81.24 | 72.78 | 78.14 | 37.55 | 80.00 | 42.94 | 43.64 | 43.70 | 51.11 | 33.52 | 23.92 | 40.19 | 38.13 | 45.05 | 73.27 | 52.34 |
| DPN | | 81.81 | 72.87 | 79.09 | 39.61 | 81.25 | 44.66 | 46.21 | 45.42 | 52.40 | 34.28 | 25.64 | 40.89 | 39.15 | 47.66 | 76.40 | 53.82 |
| POEM | TIN-597 | 78.88 | 70.32 | 74.68 | 38.59 | 77.95 | 43.41 | 44.27 | 43.00 | 50.22 | 34.64 | 25.37 | 43.09 | 37.28 | 45.43 | 81.55 | 52.58 |
| WOODS | | 81.14 | 72.39 | 76.49 | 38.50 | 79.00 | 43.23 | 44.59 | 42.37 | 49.21 | 33.03 | 24.32 | 40.23 | 39.01 | 41.19 | 47.80 | 50.17 |
| SCONE | | 80.90 | 72.30 | 77.19 | 37.91 | 78.70 | 42.36 | 43.99 | 42.81 | 49.49 | 32.19 | 24.00 | 39.26 | 37.96 | 41.80 | 48.34 | 49.95 |
| DUL (Ours) | | 77.01 | 67.67 | 74.16 | 36.16 | 78.35 | 40.79 | 41.32 | 40.17 | 46.33 | 31.42 | 22.87 | 40.79 | 35.34 | 42.74 | 45.05 | 48.01 |

Table 12: Classification error rate comparison on ImageNet-C. Here we test compared methods on a subset of the original ImageNet-C consisting of 200 classes. ID dataset is ImageNet-200.

| Method | $\mathcal{P}^{\text{SEM}}_{\text{train}}$ | Gauss. | Shot | Impul. | Defoc. | Glass | Motion | Zoom | Snow | Frost | Fog | Brit. | Contr. | Elastic | Pixel | JPEG | Avg. |
|---|---|---|---|---|---|---|---|---|---|---|---|---|---|---|---|---|---|
| MSP | | 52.2 | 70.7 | 715 | 56.0 | 55.5 | 52.6 | 54.1 | 67.3 | 67.0 | 63.5 | 56.5 | 53.3 | 46.3 | 50.3 | 51.9 | 57.9 |
| EBM (pretrain) | None | 52.2 | 70.7 | 715 | 56.0 | 55.5 | 52.6 | 54.1 | 67.3 | 67.0 | 63.5 | 56.5 | 53.3 | 46.3 | 50.3 | 51.9 | 57.9 |
| Maxlogits | | 52.2 | 70.7 | 715 | 56.0 | 55.5 | 52.6 | 54.1 | 67.3 | 67.0 | 63.5 | 56.5 | 53.3 | 46.3 | 50.3 | 51.9 | 57.9 |
| Entropy | | 51.3 | 71.2 | 71.7 | 54.4 | 54.6 | 51.8 | 53.2 | 66.9 | 66.2 | 62.7 | 55.8 | 51.4 | 45.1 | 49.7 | 51.1 | 57.1 |
| EBM (finetune) | | 52.9 | 72.2 | 72.8 | 56.0 | 56.2 | 53.5 | 54.1 | 67.8 | 67.4 | 64.0 | 57.0 | 52.2 | 46.5 | 51.4 | 52.9 | 58.5 |
| DPN | ImageNet-800 | 51.9 | 69.2 | 69.6 | 56.5 | 55.0 | 52.0 | 53.1 | 65.5 | 65.3 | 62.3 | 55.5 | 54.5 | 46.0 | 49.7 | 51.4 | 57.2 |
| WOODS | | 51.4 | 69.4 | 70.0 | 55.2 | 54.8 | 51.9 | 52.9 | 66.2 | 66.0 | 62.5 | 55.6 | 52.6 | 45.6 | 49.7 | 51.3 | 57.0 |
| SCONE | | 51.6 | 69.4 | 70.0 | 55.4 | 55.0 | 52.1 | 53.1 | 66.3 | 66.0 | 62.6 | 55.7 | 53.0 | 45.8 | 49.9 | 51.4 | 57.1 |
| DUL (Ours) | | 51.1 | 69.1 | 70.5 | 55.1 | 54.5 | 51.6 | 52.4 | 66.2 | 65.9 | 62.6 | 55.7 | 53.0 | 45.4 | 49.4 | 50.9 | 56.9 |

Table 13: Comprehensive comparison involving 15 different types of corruption from commonly-used domain adaption benchmark [52]. Substantial ($\geq 0.5$) improvement and degradation compared to the baseline MSP [6] are highlighted in blue or red respectively. DUL is the only method that achieves SOTA OOD detection performance without sacrificing generalization i.e., the value of the entire row is almost black or blue. The **best** or second best results are highlighted in bold or underline. MD is the shorthand of Mahalanobis.

| Method | $\mathcal{P}^{\text{ID}}_{\text{train}}$ | $\mathcal{P}^{\text{SEM}}_{\text{train}}$ | Gauss. | Shot | Impul. | Defoc. | Glass | Motion | Zoom | Snow | Frost | Fog | Brit. | Contr. | Elast. | Pixel | JPEG | Avg. | FPR↓ | AUC↑ |
|---|---|---|---|---|---|---|---|---|---|---|---|---|---|---|---|---|---|---|---|---|
| MSP | | | 77.0 | 67.7 | 74.5 | 36.1 | 78.5 | 40.7 | 41.4 | 40.5 | 46.7 | 31.4 | 22.6 | 40.3 | 35.3 | 42.8 | 44.9 | 48.0 | 42.0 | 89.3 |
| EBM | | None | 77.0 | 67.7 | 74.5 | 36.1 | 78.5 | 40.7 | 41.4 | 40.5 | 46.7 | 31.4 | 22.6 | 40.3 | 35.3 | 42.8 | 44.9 | 48.0 | 32.5 | 89.3 |
| Maxlogits | | | 77.0 | 67.7 | 74.5 | 36.1 | 78.5 | 40.7 | 41.4 | 40.5 | 46.7 | 31.4 | 22.6 | 40.3 | 35.3 | 42.8 | 44.9 | 48.0 | 32.9 | 89.3 |
| MD | | | 77.0 | 67.7 | 74.5 | 36.1 | 78.5 | 40.7 | 41.4 | 40.5 | 46.7 | 31.4 | 22.6 | 40.3 | 35.3 | 42.8 | 44.9 | 48.0 | 32.5 | 93.9 |
| Entropy | | | 81.7 | 72.9 | 87.0 | 38.2 | 88.9 | 42.2 | 44.6 | 45.5 | 53.9 | 33.1 | 24.4 | 39.8 | 37.2 | 47.6 | 48.8 | 52.4 | 6.6 | 98.7 |
| EBM (FT) | | | 82.3 | 73.5 | 90.6 | 39.2 | 90.7 | 41.4 | 44.4 | 47.3 | 54.9 | 34.3 | 24.7 | 39.2 | 37.7 | 51.5 | 49.3 | 53.4 | 3.6 | 98.4 |
| DPN | | | 77.6 | 68.6 | 83.6 | 39.7 | 86.8 | 43.3 | 44.9 | 47.0 | 53.7 | 36.4 | 26.1 | 43.5 | 37.6 | 47.5 | 44.5 | 52.0 | 4.3 | 98.5 |
| POEM | CIFAR-10 | ImageNet-RC | 83.7 | 76.6 | 88.0 | 44.1 | 90.1 | 44.3 | 46.4 | 52.8 | 60.8 | 39.5 | 28.9 | 43.4 | 42.5 | 56.1 | 54.5 | 56.8 | 3.3 | 99.0 |
| WOODS | | | 77.5 | 68.5 | 77.5 | 36.4 | 80.6 | 40.9 | 41.9 | 41.3 | 47.1 | 31.4 | 23.3 | 39.7 | 36.2 | 42.9 | 46.2 | 48.8 | 7.1 | 98.5 |
| SCONE | | | 76.4 | 67.3 | 75.3 | 36.2 | 77.0 | 40.9 | 41.6 | 40.2 | 45.5 | 31.4 | 23.1 | 40.1 | 35.9 | 42.4 | 45.4 | 47.9 | 7.0 | 98.5 |
| DUL (Ours) | | | 77.2 | 68.0 | 75.1 | 35.8 | 78.5 | 39.7 | 40.6 | 39.8 | 46.1 | 31.0 | 22.4 | 39.2 | 34.8 | 43.0 | 44.6 | 47.7 | 5.9 | 98.5 |
| Entropy | | | 83.3 | 75.4 | 79.8 | 38.0 | 82.7 | 42.2 | 44.2 | 44.5 | 53.4 | 32.8 | 23.7 | 38.0 | 38.1 | 45.1 | 62.9 | 52.2 | 11.6 | 97.9 |
| EBM (FT) | | | 81.2 | 72.8 | 78.1 | 37.6 | 80.0 | 42.9 | 43.6 | 43.7 | 51.1 | 33.5 | 23.9 | 40.2 | 38.1 | 45.1 | 73.3 | 52.3 | 19.4 | 87.5 |
| DPN | | | 81.8 | 72.9 | 79.1 | 39.6 | 81.3 | 44.9 | 46.2 | 45.4 | 52.4 | 34.3 | 25.6 | 40.9 | 39.2 | 47.7 | 76.4 | 53.8 | 17.3 | 94.9 |
| POEM | | TIN-597 | 78.9 | 70.3 | 74.7 | 38.6 | 78.0 | 43.4 | 44.3 | 43.0 | 50.2 | 34.6 | 25.4 | 43.1 | 37.3 | 45.4 | 81.6 | 52.6 | 34.3 | 86.8 |
| WOODS | | | 81.1 | 72.4 | 76.5 | 38.6 | 79.0 | 43.2 | 44.6 | 42.4 | 49.2 | 33.0 | 24.3 | 40.2 | 39.0 | 41.2 | 47.8 | 50.2 | 7.6 | 98.3 |
| SCONE | | | 80.9 | 72.3 | 77.2 | 37.9 | 78.7 | 42.4 | 43.4 | 42.8 | 49.5 | 32.2 | 24.0 | 39.3 | 38.0 | 41.8 | 48.3 | 50.0 | 8.0 | 98.2 |
| DUL (Ours) | | | 77.0 | 67.7 | 74.2 | 36.2 | 78.4 | 40.8 | 41.3 | 40.2 | 46.3 | 31.4 | 22.9 | 40.8 | 35.3 | 42.7 | 45.1 | 48.0 | 6.9 | 98.2 |
| MSP | | | 52.2 | 70.7 | 71.5 | 56.0 | 55.5 | 52.6 | 54.1 | 67.3 | 67.0 | 63.5 | 56.5 | 53.3 | 46.3 | 50.3 | 51.9 | 57.9 | 58.2 | 82.3 |
| EBM | | None | 52.2 | 70.7 | 71.5 | 56.0 | 55.5 | 52.6 | 54.1 | 67.3 | 67.0 | 63.5 | 56.5 | 53.3 | 46.3 | 50.3 | 51.9 | 57.9 | 32.5 | 89.3 |
| Maxlogits | | | 52.2 | 70.7 | 71.5 | 56.0 | 55.5 | 52.6 | 54.1 | 67.3 | 67.0 | 63.5 | 56.5 | 53.3 | 46.3 | 50.3 | 51.9 | 57.9 | 51.9 | 88.2 |
| Entropy | ImageNet-200 | | 51.3 | 71.2 | 71.7 | 54.4 | 54.6 | 51.8 | 53.2 | 66.2 | 62.7 | 55.8 | 45.1 | 49.7 | 51.1 | 57.1 | | | 51.1 | 89.1 |
| EBM (FT) | | | 52.9 | 72.2 | 72.8 | 56.0 | 56.2 | 53.5 | 54.1 | 67.8 | 67.4 | 64.0 | 57.0 | 52.2 | 46.5 | 51.4 | 52.9 | 58.5 | 59.7 | 87.5 |
| DPN | | ImageNet-800 | 51.9 | 69.2 | 69.6 | 56.5 | 55.6 | 52.4 | 53.1 | 65.5 | 65.0 | 62.2 | 55.5 | 54.5 | 46.0 | 49.7 | 51.4 | 57.2 | 63.8 | 87.2 |
| WOODS | | | 51.4 | 69.4 | 70.0 | 55.2 | 54.8 | 51.9 | 52.9 | 66.2 | 66.0 | 62.5 | 55.6 | 52.6 | 45.6 | 49.7 | 51.3 | 57.0 | 51.7 | 88.3 |
| SCONE | | | 51.6 | 69.4 | 70.0 | 55.4 | 55.0 | 52.1 | 53.1 | 66.2 | 66.0 | 62.6 | 55.7 | 53.0 | 45.8 | 49.9 | 51.4 | 57.1 | 52.5 | 88.2 |
| DUL (Ours) | | | 51.0 | 69.1 | 70.5 | 55.1 | 54.5 | 51.5 | 52.4 | 66.2 | 65.9 | 62.6 | 55.7 | 53.0 | 45.4 | 49.4 | 50.9 | 56.9 | 49.1 | 89.3 |

## C.6 Empirical evidence

Here we provide empirical supports to our intuition about energy-based OOD detection regularization. We calculate the entropy of predicted distribution before and after finetuning with Energy regularization [8]. The results show can support our claim in Section 4.

Table 14: Predictive entropy of predicted distribution on covariate-shifted OOD dataset before and after finetuning with energy-based OOD detection regularization [8].

| $\mathcal{P}^{\text{ID}}$ | $\mathcal{P}^{\text{SEM}}_{\text{train}}$ | Before | After |
|---|---|---|---|
| CIFAR10 | ImageNet-RC | 0.11 | 1.05 |
| CIFAR10 | TIN-597 | 0.11 | 0.14 |
| CIFAR100 | ImageNet-RC | 0.92 | 3.39 |
| CIFAR100 | TIN-597 | 0.92 | 1.15 |

# D Discussions

## D.1 Math derivation

**Differential entropy.** The proposed DUL calculates differential entropy as OOD detection measurement. Here we detail how to calculate the differential entropy of a Dirichlet distribution. The following derivation of differential entropy is taken from [9]. The differential entropy of a Dirichlet parameterized by $\alpha$ is calculated by

$$
\begin{aligned}
h[p(\mu|x)] &= -\int_S p(\mu|x)\ln(p(\mu|x))d\mu \\
&= \sum_k^K \ln\Gamma(\alpha_k) - \ln\Gamma(\alpha_0) - \sum_k^K (\alpha_k - 1)\cdot(\psi(\alpha_k) - \psi(\alpha_0))
\end{aligned}
\tag{30}
$$

where $\alpha_0$ is the strength of Dirichlet, i.e., $\alpha_0 = \sum_K \alpha_k$. $\alpha_k$ denotes the $k$-th element in $\alpha$. $\Gamma$ is the Gamma function and $\psi$ is the digamma function. Here we provide a PyTorch implementation on how to calculate distribution uncertainty measured by differential entropy.

```python
def diff_entropy(alphas):
    alpha0 = torch.sum(alphas, dim=1)
    return torch.sum(
            torch.lgamma(alphas)-(alphas-1)*(torch.digamma(alphas)-
                                            torch.digamma(alpha0).
                                            unsqueeze(1)),
            dim=1) - torch.lgamma(alpha0)

logits = model(x)
alpha = torch.Relu(logits)+1
diff_entropy = diff_entropy(alpha)
```

We refer interested readers to [9] and Gal's PhD Thesis [59] for more detailed math derivations.

## D.2 Discussion about Disparity Discrepancy

In section 4, we claim that a limited disparity discrepancy between test-time semantic OOD and covariate-shifted OOD is practical. Here we provide some empirical evidence and discussion to support such a claim.

**The key challenge of OOD detection lies in identifying ID-like semantic OOD.** As mentioned in recent works [38], effectively distinguishing between the most challenging OOD samples that are much like in-distribution (ID) data is the core challenge of OOD detection. Recent works regularize models on ID-like OOD to enhance the OOD detection performance. Since the ID-like semantic OOD samples are more difficult to be detected and more informative. For example, NTOM [17] and

POEM [11] utilizes greedy and Thompson sampling strategies to find semantic OOD samples which are more closely to ID. [38] proposes to explicitly discover outliers near ID by prompt learning.

**Semantic OOD and covariate OOD can be very similar in practice.** As shown in Fig. 4. There exists many similar samples from semantic OOD and covariate OOD in large-scale commonly used benchmarks. We borrow some examples from recent works [47] to show case.

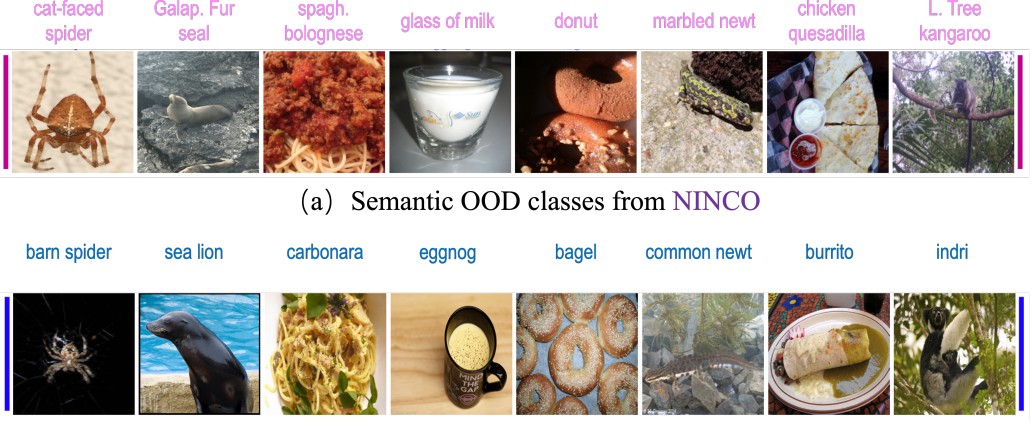

(a)  Semantic OOD classes from NINCO

(b)  ID classes from ImageNet

Figure 4: Semantic OOD samples can be very similar to ID.

## D.3  Social Impact

AI safety and trustworthiness are closely related to our work. This paper presents work to harmonize the conflicts between out-of-distribution detection methods and model generalization. The proposed method puts effort to enhance machine learning models for their safely deployment on out-of-distribution data, avoiding both undesirable behavior and degraded performance in challenging high-stake tasks. However, due to the bias from data used by current OOD detection benchmark, e.g., large-scale ImageNet, the ones using the proposed method need to carefully consider the selection of auxiliary outliers for safety-critical applications.

