# OpenReview forum: "The Best of Both Worlds: On the Dilemma of Out-of-distribution Detection"
_NeurIPS.cc/2024/Conference — NeurIPS 2024 poster_

### Official Review · Reviewer_DUjY · 2024-07-08

**Soundness:** 3
**Presentation:** 4
**Contribution:** 4
**Rating:** 6
**Confidence:** 4

**Summary:**

This paper proposes a novel method for finetuning a classifier for both OOD generalization and OOD detection. The paper presents a novel objective function for OOD generalization and detection based on the Bayesian framework and OOD generalization theory. Extensive experimental results suggest that the proposed method is effective.

**Strengths:**

* The paper addresses an important problem of achieving OOD generalization and OOD detection simultaneously. This task has high significance regarding the safety of ML systems.
* Overall, the exposition of the paper is clear and easy to follow.
* The paper presents a sound theoretical analysis on why some existing OOD detection approaches have failed in OOD generalization.
* The experiments are conducted in a reasonable setting, and the results are promising.

**Weaknesses:**

* It is possible that I missed it, but the paper does not provide a direct theoretical argument supporting the main objective function (Eq. (11)). The most important term in Eq. (11) is the constraint that the entropy of p(y|x) should not change before and after the fine-tuning. This term maintains OOD generalization, but the justification is weak.
* Related to the above point, the statement in Lines 244-245 ("While generalization is related to 245 the overall uncertainty of p(yˆ|x) as we mentioned in related works (both AU and DU).") argues that this entropy (i.e., "overall uncertainty") is connected to the generalization performance. Still, I could not find the supporting information in the related work section.

**Questions:**

* Is there a reason why we should enforce the "non-increased overall uncertainty" constraint (the constraint in Eq. (11)) only on semantic OOD data? For example, what if we enforce this on the in-distribution dataset? Is there a theoretical justification for this choice, or is it more empirical?

**Limitations:**

* The paper addresses its limitation in the conclusion section. The limitation of the paper does not raise a concern.

---

> ### Author Rebuttal · Authors · 2024-08-06
>
> We thank the reviewer for recognizing the importance of our research problem, clear exposition, sound theoretical analysis, and promising results. We appreciate your support and constructive suggestions and address your concerns as follows.
>
> - **W1.** It is possible that I missed it, but the paper does not provide a direct theoretical argument supporting the main objective function (Eq. (11)). The most important term in Eq. (11) is the constraint that the entropy of p(y|x) should not change before and after the fine-tuning. This term maintains OOD generalization, but the justification is weak.
>
> Thanks for raising a concern about direct theoretical support to Eq.11. Theorem 1 can directly support the learning objective in Eq.11. In Theorem 1, the OOD detection loss is negatively correlated with OOD generation loss. Please note that the OOD detection loss we discuss here is the cross entropy between model prediction and uniform distribution proposed by [7]. This term explicitly encourages high entropy on OOD samples and is proven to be harmful to model generalization by Theorem 1. Thus in Eq.11, we disencourages high-entropy prediction, which is a natural and effective solution. More importantly, in section 5, line 230-248, we show that this constraint is not harmful to OOD detection theoretically by revisting the property of various uncertainty. We will add more discussion and smoothen the logic in the revised paper.
>
> - **W2.** Related to the above point, the statement in Lines 244-245 ("While generalization is related to the overall uncertainty of p(y|x) as we mentioned in related works (both AU and DU).") argues that this entropy (i.e., "overall uncertainty") is connected to the generalization performance. Still, I could not find the supporting information in the related work section.
>
> Thanks for your comments. Many recent works in OOD generalization literature can support our statement. Here we provide a brief discussion. The recent success of OOD generalization [A] [C] [D] [E] has shown that overall uncertainty (entropy) is closely connected to the generalization performance. By iteratively minimizing the predictive entropy, **the model classification accuracy on test data can be significantly improved and vice versa**. The initial intuition from [C] is based on the observations that models tend to be more accurate on images for which they make predictions with higher confidence. Besides, earlier works [B] in semi-supervised literature also show that decreasing predictive entropy can improve model accuracy. The natural logical extension of this observation is to enforce models to not increase their entropy on test samples, which is the key motivation of our DUL. We will elaborate on this in revision.
>
> - **Q1.** Is there a reason why we should enforce the "non-increased overall uncertainty" constraint (the constraint in Eq. (11)) only on semantic OOD data? For example, what if we enforce this on the in-distribution dataset? Is there a theoretical justification for this choice, or is it more empirical?
>
> Thanks for your interesting question. **Our choice is supported by both empirical and theoretical evidence.** The answer to your question is threefold. First, for covariant-shifted OOD. As we have shown before, many previous works in OOD generalization have demonstrated both theoretically [A, B] and empirically [C, D, E] that entropy on covariant-shifted OOD is negatively related to classification performance. Secondly, when it comes to semantic OOD, we have theoretically proved that high entropy will also result in degraded classification performance in Theorem 1. Thirdly, however, enforcing low overall uncertainty on the in-distribution dataset may be not necessary. Since during pretraining and finetuning, the standard cross-entropy loss has enforced the model to be highly confident on ID data [F]. **To summarize, we should discourage high overall uncertainty on all three types of data. However, only constraining on the semantic OOD is enough. Because in our settings, we cannot access covariant-shifted OOD and the entropy on ID has been constrained by standard CE loss.** We will add these explanations to the revised paper.
>
> [A] The Entropy Enigma: Success and Failure of Entropy Minimization. ICLR'24
>
> [B] Semi-supervised learning by entropy minimization. NIPS'04
>
> [C] Fully test-time adaptation by entropy minimization. ICLR'21
>
> [D] Memo: Test time robustness via adaptation and augmentation. NIPS'22
>
> [E] Towards stable test-time adaptation in dynamic wild world. ICLR'23
>
> [F] On calibration of modern neural networks. ICML'17

---

> > ### Comment · Reviewer_DUjY · 2024-08-11
> > **Thanks for your response**
> >
> > Thank you for your detailed response.  Overall, all my other questions are addressed well. Regarding W1, I missed that the OOD detection loss is the cross-entropy with respect to the uniform distribution. Using the same mathematical symbol for representing the same quantity could potentially improve the clarity.

---

> > > ### Author Response · Authors · 2024-08-11
> > >
> > > Thank you for your suggestion. We are pleased to hear that most of your questions have been addressed. It is our responsibility to present mathematical notations clearly and avoid ambiguity. In lines 135-136, we describe in words that $L_{reg}$ is the cross-entropy between the prediction and the uniform distribution for MSP detectors. According to your advice, we will use $H(F(x), U)$ instead, where U denotes the uniform distribution and $H(\cdot , \cdot)$ represents the cross-entropy. This notation is consistent with the rest of this paper. We will also carefully revise the relative parts.
> > >
> > > If there are any additional questions that you would like to discuss with us, please feel free to post, we will continuously work on them and actively address your concerns.

---

### Official Review · Reviewer_Tcyq · 2024-07-11

**Soundness:** 3
**Presentation:** 4
**Contribution:** 4
**Rating:** 8
**Confidence:** 4

**Summary:**

This paper theoretically characterizes the underlying dilemma in SOTA OOD detection method.  The authors find that OOD detection performance of state-of-the-art methods is achieved with tradeoff between OOD detection and classification. Accordingly, the authors provide an uncertainty-based strategy which decouples these two tasks and thus addresses the dilemma. Plenty of experiments are conducted which well support the theory and validate the effectiveness of the proposed model.

**Strengths:**

1. The findings of this paper are very interesting. This work studies the dilemma between OOD detection and classification generalization. For the first time, the authors provide a theoretical perspective which well reveals the underlying tradeoff. This theoretical result provides strict and intuitive explanations for this phenomenon, which is important and enlightening.
2. The proposed uncertainty-decoupled strategy is reasonable for the dilemma, and the paper is technically solid. The proposed method is developed according to the induced theory which validates the potential of the theory in inspiring other models.
3. The organization and writing of this paper are clear, making the contributions easy to catch. The experiments sufficiently validate both the theory and the proposed method.

**Weaknesses:**

I am not sure are there any existing works implicitly considering decoupling these two tasks, i.e., OOD detection and generalization. Please provide more clarifications.

The theory is very promising. So it will be better if the authors could provide more discussion on other potential strategies that could also addressing the dilemma. I think it will be very useful for readers.

**Questions:**

1. Figure 1 is very helpful and intuitive, while it seems not clear - which dataset the results are reported in the right figure.
2. The authors should explain more about the “tradeoff area”. Although it seems ok, but I think more detailed description is necessary.
3. “Reducing the dependency on auxiliary OOD data can be an interesting research direction for the future exploration.” I think it is interesting to reduce the dependency on auxiliary OOD data. However, could the authors provide some potential directions?

**Limitations:**

See Weaknesses

---

> ### Author Rebuttal · Authors · 2024-08-06
>
> We sincerely thank the reviewer for your valuable comments and appreciate your recognition of the interesting findings, important and enlightening theoretical results as well as sufficient experiments. We provide detailed responses to address your concerns.
>
> - **W1.** I am not sure are there any existing works implicitly considering decoupling these two tasks, i.e., OOD detection and generalization. Please provide more clarifications.
>
> Thanks for your suggestions. Here we provide a brief survey of existing works related to both these two tasks. [A] considers both OOD detection and generalization in vision-language models (i.e., CLIP), they propose to hierarchically construct the text description for a certain category and enhance ID classification and OOD detection. [B] utilizes the diffusion model to generate virtual OOD samples which can be used for enhancing OOD detection and generalization simultaneously. [C] focus on improving OOD generalization performance in a realistic open-set setting, which is capable of simultaneously handling covariate-shifted OOD data and detecting semantic OOD data. Though there exist a few works that pursue these two targets altogether, the instinctive relationship between them is notably unexplored, not to mention an ideal solution. Thus we believe our findings is distinguished from these works. We will add more detailed clarifications in the revision according to your suggestion.
>
> [A] Category-Extensible Out-of-Distribution Detection via Hierarchical Context Descriptions. NIPS'23
>
> [B] Dream the Impossible: Outlier Imagination with Diffusion Models. NIPS'23
>
> [C] ATTA: anomaly-aware test-time adaptation for out-of-distribution detection in segmentation. NIPS'23
>
> - **W2.** The theory is very promising. So it would be better if the authors could provide more discussion on other potential strategies that could also address the dilemma. I think it will be very useful for readers.
>
> Thanks for your suggestion. According to our analysis, previous OOD detection methods explicitly/implicitly encourage high-entropy prediction resulting in the dilemma. The most straightforward way to address this limitation is enforcing unchanged entropy as we devised in this paper. Besides, we notice that many recent OOD generalization works utilized unsupervised entropy minimization loss to further enhance the classification performance [28, 29, 30]. Their core idea is coherent with our DUL. It is worth trying new strategies such as conducting entropy minimization on unlabeled test data in the future.
>
> [28] Tent: Fully test-time adaptation by entropy minimization. ICLR'21
>
> [29] Memo: Test time robustness via adaptation and augmentation. NIPS'22
>
> [30] Towards stable test-time adaptation in dynamic wild world. ICLR'23
>
> - **Q1.** Figure 1 is very helpful and intuitive, while it seems not clear - which dataset the results are reported in the right figure?
>
> We visualize the results on CIFAR-100 when TIN-597 serves as auxiliary OOD data in Figure 1 (b). Consist with the tabular results in Table 1. We will clearly explain this in revision.
>
> - **Q2.** The authors should explain more about the "tradeoff area". Although it seems ok, I think a more detailed description is necessary.
>
> Thanks for your suggestion. The trade-off area in Fig. 1 (b) denotes the area where the investigated OOD detection methods perform better than the baseline MSP (the pretrained model without any OOD detection regularization) but yield a higher error rate compared to MSP. These methods exhibit better OOD detection performance but sacrifice OOD generalization ability. Thus we say that they lie in a "trade-off area". We will clearly explain this in the revised paper.
>
> - **Q3.** I think it is interesting to reduce the dependency on auxiliary OOD data. However, could the authors provide some potential directions?
>
> Sure. As far as we can tell, there exist three directions in this literature. I. **Outlier sampling strategies** that choose the most informative outliers for model regularization. For example, greedy sampling[17] and Thompson sampling[11]. These methods are of better data efficiency compared to others. II. **Leveraging external knowledge from pretrain models**. Recent [34] utilizes the diffusion model to generate virtual outliers which can remove the dependency on real auxiliary OOD data. III. **Directly devising OOD detection methods without auxiliary OOD data.** Many works aim to enhance OOD detection performance given only a pretrain model. Though their practical performance is often sub-optimal compared to the counterparts explicitly regularized on OOD data, they are usually easier to deploy in applications.

---

> > ### Comment · Reviewer_Tcyq · 2024-08-12
> >
> > Thank you for your detailed rebuttal. The author addressed my problem so well that I kept my rating on the work.

---

### Official Review · Reviewer_31hb · 2024-07-12

**Soundness:** 3
**Presentation:** 3
**Contribution:** 3
**Rating:** 6
**Confidence:** 5

**Summary:**

This paper reveals the trade-off dilemma of OOD detection and OOD generalization for current SOTA OOD detectors from both theoretical and empirical perspectives. The authors employ a transfer learning framework to analyze the generalization error bound for MSP-based OOD detectors and identify the optimization conflicts between the training objectives of OOD detection and OOD generalization. They find similar patterns for Energy-based detectors. From the theoretical analysis, this paper derives a decoupled uncertainty learning technique for the dual-optimal performance for OOD detection and generalization. Experiments on CIFAR10/100 and ImageNet-200 validate its effectiveness.

**Strengths:**

1. The research problem is valuable and well-presented.
2. The theoretical analysis for the objective conflicts for MSP- and Energy-based OOD detectors is sound.
3. The experiments show promising performance for the proposed DUL technique, which successfully maintains the OOD generalization capability while achieving competitive OOD detection results.

**Weaknesses:**

1. This paper only discusses the MSP- and Energy-based OOD detectors, which adopt exactly the same outputs (i.e., logits) from a neural network to perform ID classification (and OOD generalization) and OOD detection, where the optimization conflict is intuitive. However, other superior OOD detectors employ extra output branches aside from the classification logits (even if they share the same backbone for feature extraction) for OOD detection, such as [1][2][3], which shows promising robustness for covariate shifts[2]. Is the theoretical analysis and DUL technique still applicable to those types of OOD detectors?
2. As the authors claim,  DUL can lead to a dual-optimal performance for both OOD detection and OOD generalization. However, according to Table 1, DUL does not always win first place across different experimental settings. For instance, DUL and $\dagger$ DUL are surpassed by POEM on OOD detection performance by a non-negligible margin in the CIFAR10/100+ImageNet-RC setting. It seems DUL just achieves a better trade-off rather than the dual-optimal results. A proper explanation should be given.
3. DUL seems to basically add a normalization/regularization on the model's output logits with knowledge (measured by Bayesian framework) distilled from a pre-trained ID classifier. It limits the application and novelty of this paper. What happens if the model is trained from scratch?
4. Besides, adding Gaussian noise is a too trivial setting to evaluate the robustness of covariate shifts. How about evaluating OOD detectors in standard domain adaptation settings?

[1] J Bitterwolf et al. “Breaking Down Out-of-Distribution Detection: Many Methods Based on OOD Training Data Estimate a Combination of the Same Core Quantities”. ICML 2022.

[2] L Kai et al. "Category-Extensible Out-of-Distribution Detection via Hierarchical Context Descriptions". NeurIPS 2023.

[3] W Miao et al. "Out-of-Distribution Detection in Long-Tailed Recognition with Calibrated Outlier Class Learning". AAAI 2024.

**Questions:**

Although OOD detection and OOD generalization are both frequently discussed in the literature, it is still weird when they come up together. It is basically because the D(istribution) for OOD detection and OOD generalization has different meanings. For OOD detection, it means the collection of label y. For OOD generalization, it means the data distribution for sample x. A proper clarification or modification can be taken into consideration

**Limitations:**

Please see weakness.

---

> ### Author Rebuttal · Authors · 2024-08-06
>
> We sincerely thank the reviewer for your valuable comments and appreciate your recognition of the clear presentation and effective method. We provide detailed responses to the constructive comments.
>
> - **W1.** The paper only discusses the MSP- and Energy-based OOD detectors, which are both adapted to logits from a neural network. There exist superior OOD detectors that use extra output branches aside from the classification logits for OOD detection (e.g. [1] [2] [3]) which shows promising robustness for covariate shifts[2]. Is the theoretical analysis and DUL technique still applicable to these OOD detectors?
>
> Thanks for your insightful suggestions. According to your advice, we conduct additional experiments on OOD detectors with the K+1 class branch. First, the observed dilemma also exists in these types of OOD detectors. **We would like to acknowledge that, our theoretical analysis is not directly applicable for OOD detectors with K+1 class for now. However, the DUL technique is still applicable.** The training strategy we suggest for the K+1 classifier is: 1) First train a standard K-class classifier on the ID dataset. 2) Then replace the last fully connect layer to K+1 (random initialize its parameters). 3) During finetuning, one can obtain the predicted distribution $p^{k\leq K}(\hat{y}|\tilde{x})$ of the first K class by normalizing their logits $[f_1,\dots,f_k]$ with softmax. Then explicitly penalize high entropy on these ID classes as in Eq.12. The loss function is CE loss and $(f_{k+1}(x)-m_{in})^2+(f_{k+1}(\tilde{x})-m_{out})^2+KL(p^{k\leq K}(\hat{y}|\tilde{x})||p_0^{k\leq K}(\hat{y}|\tilde{x}))$. The results show that the DUL technique is potentially useful for detectors with an additional branch. **We promise to make the limitation transparent in the revision.** Further analysis from a feature learning perspective may be needed in future work.
>
> |         |OOD-ACC|FPR|AUROC|
> | -------------------- | -------| -----| ----- |
> |MSP|88.11|37.04|90.91|
> |K+1 classifier|84.77|3.22|99.11|
> |K+1 classifer w. DUL|87.99|7.12|98.35|
>
> Besides, please kindly remind that DUL is devised in a finetune manner. Compared to K+1 classifier that requires re-training the classifier from scratch, DUL can be applied to any pre-trained model (e.g., torchvision, huggingface), with modest compute overhead ($\leq$ 20 epochs).
>
> - **W2.** The authors claim that DUL can lead to dual-optimal performance for both OOD detection and generalization. However, DUL is surpassed by POEM on OOD detection in the CIFAR10/100+ImageNet-RC setting. A proper explanation should be given.
>
> Thanks for the comments. Our DUL is initially devised in a finetune manner for computationally effectiveness. **DUL is finetuned on ImageNet-RC for only 20 epochs**. However, the official implementation of **POEM is trained from scratch for 200 epochs**. Thus it is reasonable that DUL does not outperform POEM in a few certain settings. According to your advice, we finetune DUL in CIFAR10/ImageNet-RC setting for more epochs (100). We observe that **DUL achieves even better OOD detection performance compared to POEM on CIFAR10/ImageNet-RC with longer training.**
>
> |             | OOD-ACC | FPR  | AUROC |      |
> | ----------------- | ------- | ---- | ----- | ---- |
> | POEM (200 epochs) | 78.89   | 3.32 | 98.99 |      |
> | DUL (100 epochs)  | 88.13   | 2.75 | 98.06 |      |
>
> - **W3.** DUL seems to basically add a normalization/regularization on the model's output logits with knowledge (measured by Bayesian framework) distilled from a pre-trained ID classifier. It limits the application and novelty of this paper. What happens if the model is trained from scratch?
>
> Thanks for the interesting question and careful reading. Initially, we devise DUL in a finetune manner following EBM [4] for **effectiveness**. Thus it can be applied with any pre-trained model. **If the model is trained from scratch, one can simply introduce a two-stage training strategy.** First, train an ID classifier, and then finetune on OOD data with our DUL. It would take only 20 additional epochs for the second stage finetuning to get the superior performance we reported.
>
> - **W4.** Besides, adding Gaussian noise is a too trivial setting to evaluate the robustness of covariate shifts. How about evaluating OOD detectors in standard domain adaptation settings?
>
> Thanks for your valuable suggestions. Please kindly remind that we have provided a comprehensive evaluation involving 15 different types of corruption (e.g., snow, rainy, defocus...) on CIFAR-C/ImageNet-C [52] in Appendix C (Tab. 8, 9, and 10). To our best knowledge, CIFAR10/100-C and ImageNet-C are widely used in domain adaption settings. **We have reorganized the results into Table 1, global response PDF. The overall performance of DUL consistently outperforms its counterparts.**
>
> - **Q1.** Although OOD detection and OOD generalization are both frequently discussed in the literature, it is still weird when they come up together. It is basically because the D(istribution) for OOD detection and OOD generalization has different meanings. For OOD detection, it means the collection of label y. For OOD generalization, it means the data distribution for sample x. A proper clarification or modification can be taken into consideration.
>
> Thanks for your suggestion. In the classification context, distribution shift means the joint distribution $p(x,y)$ is different during training and testing. In this paper, OOD detection targets semantic OOD where the label $y$ of which do not belong to any known classes. OOD generalization aims to properly classify data where the $x$ undergoes changes in outlook or shift in style but still belongs to known classes. **These types of data can organically arise when deploying models in the open world [4]**. Thus it deserves to find out how to handle them in a unified framework. We will further clarify the definition of different types of OOD samples in an open-world setting according to your suggestion.

---

> > ### Comment · Reviewer_31hb · 2024-08-11
> >
> > I appreciated the author's responses and have raised my score to 6. It is highly recommended to add those discussions in the revised version.

---

> > > ### Author Response · Authors · 2024-08-11
> > >
> > > Thanks for your positive feedback! We will carefully revise the related parts according to your suggestion.

---

### Official Review · Reviewer_gJee · 2024-07-12

**Soundness:** 3
**Presentation:** 3
**Contribution:** 3
**Rating:** 5
**Confidence:** 4

**Summary:**

This paper addresses out-of-distribution (OOD) detection and generalization problems. The authors show the sensitive-robust dilemma in learning objectives of OOD detection and generalization and propose a decoupled uncertainty learning (DUL) method to harmonize the above conflict. The proposed method only encourages high distribution uncertainty on OOD data and explicitly discourages high entropy in the final prediction. Experiments on some datasets show the DUL method achieves better performance compared to several baselines.

**Strengths:**

1. Provide a detailed analysis of the sensitive-robust dilemma between OOD detection and generalization by leveraging transfer learning theory.

2. Develop a decoupled uncertainty learning algorithm that explicitly discourages high entropy in the final prediction to keep the OOD generalization ability of the model.

3. Experiments on benchmarks demonstrate the decoupled uncertainty learning objective achieves better performance compared to baselines.

**Weaknesses:**

1. While analyzing the sensitive-robust dilemma via transfer learning theory is novel, the phenomenon that existing SOTA OOD detection methods may suffer from catastrophic degradation in terms of OOD generalization has been reported in [4].

2. The decoupled uncertainty learning objective introduces an additional regularization term named unchanged overall uncertainty. It is difficult to understand why DUL even performs better than Entropy [7] and EBM (finetune) [8] on OOD detection tasks.

3. Only simple Gaussian noise is used to reflect the OOD generalization ability, while there are many types of out-of-distributions like corruptions. Therefore, the results in Table 1 can not represent the performance of evaluated methods because different kinds of OOD data often behave differently under the same model [A].

[A] OoD-Bench: Quantifying and Understanding Two Dimensions of Out-of-Distribution Generalization, CVPR 2022.

**Questions:**

1. According to Eq.9, if keep the overall uncertainty and enlarge the distributional uncertainty (for OOD detection), the data uncertainty must be reduced. Could the authors provide the visualization of data uncertainty?

2. The reviewer suggests two more baselines. (1) In the original Entropy [7] or EBM (finetune) [8], reduce the weight of outlier regularization terms and show the performance on both OOD detection and generalization. (2) Add the unchanged overall uncertainty term in Eq.12 to the original Entropy [7] or EBM (finetune) [8], and report the performance. Those two baselines would help to demonstrate the effectiveness of the bayesian framework in decoupled uncertainty learning.

**Limitations:**

The authors adequately addressed the limitations.

---

> ### Author Rebuttal · Authors · 2024-08-06
>
> We thank the reviewer for the thoughtful and thorough comments on our paper and for recognizing the contribution of our theoretical analysis and the superiority of our DUL method over current SoTA methods.
>
> - **W1.** While analyzing the sensitive-robust dilemma via transfer learning theory is novel, the conflict between OOD detection and generalization has been reported in [4].
>
> Thanks for mentioning [4]. We are pleased to acknowledge that the conflict between these two tasks is first reported in [4]. Our contribution lies in theoretically demystifying the underlying reasons and providing a theory-inspired solution. We will further clarify the difference and highlight this point in the revision.
>
> - **W2.** DUL introduces an additional regularization. It is difficult to understand why DUL even performs better than Entropy and EBM on OOD detection tasks.
>
> Although Entropy and EBM are specified for OOD detection. Their techniques may suffer from potential pitfalls. **Entropy uses MSP for OOD detection which has been proven to be overconfident and sub-optimal in [8].** In contrast, our DUL uses differential entropy which enjoys a good interpretability from a Bayesian perspective. Besides, **EBM enforces the energy score of all ID samples to be less than a fixed threshold**, which can be problematic when there are abnormal ID samples in the dataset (e.g., label noise). However, in Eq.12, DUL finetunes the differential entropy on ID to be relatively higher than that before finetuning. The differential entropy threshold in DUL is set in a **data-adaptive manner**. Thus DUL can perform better than these baselines. Thanks for your insightful comments. We will further elaborate on this in the revision.
>
> - **W3.** Only Gaussian noise can not represent the performance of evaluated methods, since the same model behaves differently on various OOD data [A].
>
> Thanks for your suggestion. Please kindly remind that we have provided a comprehensive evaluation involving 15 different types of corruption (e.g., snow, rainy, defocus...) on CIFAR10/100-C and ImageNet-C [52] in Appendix C (Tab. 8, 9, and 10 in our manuscript). **We have reorganized them into Table 1 in the global response PDF**. The overall performance of our DUL consistently outperforms its counterparts.
>
> - **Q1.** According to Eq.9, if keep the overall uncertainty and enlarge the distributional uncertainty (for OOD detection), the data uncertainty must be reduced. Could the authors provide the visualization of data uncertainty?
>
> Very insightful question. According to your suggestions, **we visualize the data uncertainty on semantic OOD test data (Textures) when CIFAR10 is ID in Figure 1 in the global response PDF**. The investigated methods are 1) pretrained model training on ID dataset only, 2) finetuned model with OOD detection regularization (ablating the last term in Eq.12), and 3) finetuned model with the full DUL method described by Eq.12. **The results meet your expectation**. We will add these results to the revised paper.
>
> - **Q2.** The reviewer suggests two more baselines. (1) In the original Entropy [7] or EBM (finetune) [8], reduce the weight of outlier regularization terms and show the performance on both OOD detection and generalization. (2) Add the unchanged overall uncertainty term in Eq.12 to the original Entropy [7] or EBM (finetune) [8], and report the performance. Those two baselines would help to demonstrate the effectiveness of the Bayesian framework in decoupled uncertainty learning.
>
> According to your advice, we conduct additional experiments. See also **Table 2 and Table 3 in the global rebuttal PDF**.
>
> (1)  We tune the weight of outlier regularization term from [0,1e-4, 1e-3, 1e-2, 1e-1] for EBM [8], [0, 5e-4, 5e-3, 5e-2, 5e-1] for Entropy [7] and report the FPR (OOD detection metric) and error rate (Err, OOD generalization metric).
>
> | Entropy   |       |       |       |       |       |      | Energy    |       |       |       |       |       |
> | --------- | ----- | ----- | ----- | ----- | ----- | ---- | --------- | ----- | ----- | ----- | ----- | ----- |
> | $\lambda$ | 0     | 5e-4  | 5e-3  | 5e-2  | 5e-1  |      | $\lambda$ | 0     | 1e-4  | 1e-3  | 1e-2  | 1e-1  |
> | FPR.      | 35.15 | 8.36  | 6.37  | 5.71  | 5.60  |      | FPR.      | 20.57 | 14.69 | 13.54 | 8.15  | 6.11  |
> | Err.      | 9.55  | 13.58 | 15.48 | 17.97 | 18.53 |      | Err.      | 9.55  | 9.46  | 10.32 | 16.43 | 24.38 |
>
>
> When the regularization strength increases, OOD detection performance improves (lower FPR.), while the OOD generalization performance degrades (higher error rate).
>
> (2) We add the unchanged overall uncertainty term in Eq.12 to Entropy [7] and EBM (finetune) [8], and report the performance on CIFAR10/ImageNet-RC.
>
> | Method      | ID-Acc. | OOD-Acc. | FPR.  | AUROC |
> | ----------- | ------- | -------- | ----- | ----- |
> | EBM      | 95.93   | 81.47    | 5.58  | 97.75 |
> | EBM+reg  | 95.19   | 87.45    | 6.17  | 98.45 |
> | Entropy     | 96.04   | 72.57    | 6.63  | 98.72 |
> | Entropy+reg | 96.10   | 87.41    | 29.56 | 95.92 |
> | DUL         | 96.04   | 88.01    | 5.71  | 98.61 |
>
> The results show that DUL regularization can also benefit EBM. However, combining Entropy with our regularization can not improve the accuracy substantially. This is not surprising, since the target of Entropy (high entropy prediction) and our DUL (non-increased entropy) directly conflict according to Theorem 1 in our manuscript. Besides, our DUL generally outperforms the suggested baselines.
>
> [4] Feed Two Birds with One Scone: Exploiting Wild Data for Both Out-of-Distribution Generalization and Detection, ICML'23
>
> [8] Energy-based out-of-distribution detection. NIPS'20
>
> [52] Benchmarking neural network robustness to common corruptions and surface variations. ICLR'19

---

> > ### Comment · Reviewer_gJee · 2024-08-12
> >
> > I appreciate author's feedback. Some of my concern are resolved. I keep my original positive score.

---

> ### Author Response · Authors · 2024-08-12
>
> Thanks for your positive feedback. If there are any additional questions that you would like to discuss, please feel free to let us know, we will continuously work on them and actively address your concerns. Your previous suggestions, especially the two suggested new baselines and visualization of uncertainty, have improved even our own understanding on the proposed method. Thanks again for your review and invaluable suggestions.

---

### Author Rebuttal · Authors · 2024-08-06

Dear PCs, SACs, ACs, and Reviewers,

Thanks for your valuable feedback and insightful reviews, which greatly improved the paper. We are deeply encouraged that all the reviewers gave positive assessment on our work. This is a **clear** and **well-presented** (Reviewer 31hb, Tcyq) manuscript with a **valuable** and **important** research problem (Reviewer DUjY, 31hb) and **sound**, **strict**, **important, and enlightening** theoretical analysis (Reviewer DUjY, Tcyq, 31hb). The proposed method is **reasonable** and **novel** (Reviewer DUjY, Tcyq). The **promising** and **extensive** experimental results **can validate the effectiveness of the proposed method**. (Reviewer gJee, 31hb, DUjY, Tcyq).

In the rebuttal, we addressed the following raised concerns/misunderstandings.

- We have provided a comprehensive evaluation involving 15 diverse corruptions from commonly-used domain adaption benchmarks i.e., CIFAR10/100-C and ImageNet-C (Table 1, global response PDF).
- We have provided a visualization of various uncertainties (Figure 1 in the PDF).
- We have shown how to extend DUL to the k+1 classifier and model trained from scratch and promise to make the potential limitations transparent in revision (Table 2 in the PDF).
- We have added experiments on two baselines suggested by Reviewer gJee (Table 2, Table 3 in the PDF).
- We have carefully revised all the presentation issues throughout the paper.

We believe these clarifications and additional results strengthen our paper and address the reviewers' concerns. We understand the workload that reviewers and AC face, and appreciate the effort already put into evaluating our work. If there are any additional insights, questions, or clarifications that you would like to discuss with us, we would be very grateful to hear them, your feedback is invaluable for the improvement of our research.

Best regards,

Authors

---

### Comment · Area_Chair_MTXq · 2024-08-08
**Discussion has begun**

Dear Reviewers,

Thank you for taking the time to review the paper. The discussion has begun, and active participation is highly appreciated and recommended.

Thanks for your continued efforts and contributions to NeurIPS 2024.

Best regards,

Your Area Chair

---

### Decision · Program_Chairs · 2024-09-25

**Decision:**

Accept (poster)

**Comment:**

This paper investigated a very interesting problem: the relationship between OOD detection ability and OOD generalization ability, suggesting us be aware of purely pursuing the OOD detection ability in practice and theory. All reviewers agree that this contribution is valuable to the field.